# HARNESSING WEBPAGE UIS FOR TEXT-RICH VISUAL UNDERSTANDING

◇**Junpeng Liu,** ♡**Tianyue Ou,**\* ¶**Yifan Song,**\* ♡**Yuxiao Qu,**\*
◇**Wai Lam,** ♡**Chenyan Xiong,** ♣**Wenhu Chen,** ♡**Graham Neubig,** ♡**Xiang Yue**†

♡Carnegie Mellon University, ◇The Chinese University of Hong Kong
¶Peking University, ♣University of Waterloo
jpliu@link.cuhk.edu.hk    xyue2@andrew.cmu.edu

## ABSTRACT

Text-rich visual understanding—the ability to process environments where dense textual content is integrated with visuals—is crucial for multimodal large language models (MLLMs) to interact effectively with structured environments. To enhance this capability, we propose synthesizing general multimodal instructions from webpage UIs using text-based large language models (LLMs). Despite lacking direct visual input, text-based LLMs are able to process structured text representations from webpage accessibility trees. These instructions are then paired with UI screenshots to train multimodal models. We introduce MultiUI, a dataset containing 7.3 million samples from 1 million websites, covering diverse multimodal tasks and UI layouts. Models trained on MultiUI not only excel in web UI tasks—achieving up to a 48% improvement on VisualWebBench and a 19.1% boost in element accuracy on a web agent dataset Mind2Web—but also generalize surprisingly well to non-web UI tasks and even to non-UI domains, such as document understanding, OCR, and chart interpretation. These results highlight the broad applicability of web UI data for advancing text-rich visual understanding across various scenarios.

## 1 INTRODUCTION

*Text-rich visual understanding*, the ability to interpret environments where textual content is densely intertwined with visual elements, is a crucial cognitive skill in humans. For multimodal large language models (MLLMs) (OpenAI, 2023; Liu et al., 2024b), replicating this ability is essential for tasks that involve complex text-visual interactions, such as document processing (Mathew et al., 2021; Singh et al., 2019; Liu et al., 2023c), web navigation (Liu et al., 2024c; Deng et al., 2023), chart interpretation (Masry et al., 2022) and text-rich visual reasoning (Yue et al., 2024b). These tasks require models to integrate dense textual information with surrounding visuals, enabling AI systems to interact intelligently with the increasingly text-rich digital landscape (Koh et al., 2024).

To advance text-rich visual understanding in MLLMs, we propose leveraging *webpage UIs* as a naturally structured, diverse, and text-dense data source. Web UIs, where textual content is often central and tightly integrated with visual elements and interactivity, offer an ideal setting for training models to interpret and navigate complex text-visual interactions.

Existing approaches to using web content in multimodal models have limitations (Figure 1). Rule-based extraction of images and their surrounding text (Zhu et al., 2023; Schuhmann et al., 2021; 2022) often introduces noise and lacks contextual depth. Converting screenshots into simplified HTML structures (Lee et al., 2023; Gao et al., 2024) imposes a rigid format that limits generalization across domains. Models like GPT-4 generate captions (Chen et al., 2023) for web images but frequently overlook the rich interaction between text and visuals.

Our approach addresses these limitations by *synthesizing general multimodal instructions from webpage UIs using text-based LLMs*. Although text-based LLMs lack direct visual input, they can

---

\*Co-second author. Order determined by dice roll.
†Corresponding Author.

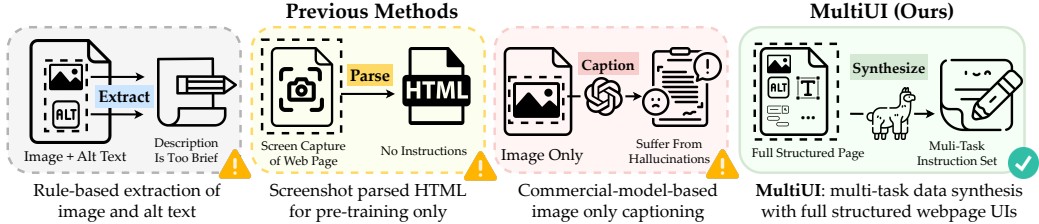

Figure 1: **MultiUI compared with previous methods.** Our proposed MultiUI construction approach synthesizes full structured webpage UIs into multimodal instruction samples of versatile tasks by harnessing powerful LLMs, which leads to more generalizable training samples.

effectively process the textual representations of webpages. By reading the cleaned accessibility tree—a structured and refined representation of a webpage's HTML and metadata—LLMs generate meaningful instructions that capture both the content and interactions present on the page. These generated instructions are then paired with UI screenshots to train multimodal models, allowing them to learn from both text and visual representations.

To facilitate this, we introduce MultiUI, an open-source dataset containing 7.3 million samples spanning 1 million websites and various visual understanding tasks. Our pipeline captures key web elements and layout structures using screenshots and enhanced accessibility trees, filtering out irrelevant data while preserving the core structure of web UIs. Our task taxonomy, which covers three categories and nine tasks, ensures that models trained on MultiUI generalize across a wide range of multimodal interactions. Additionally, we introduce variations in device types, aspect ratios, and question formats to further increase the dataset's diversity and enhance model robustness.

Our experiments show that training on MultiUI significantly improves model performance in both UI-related and general multimodal tasks. Notably, models trained on MultiUI achieved up to a 48% improvement on VisualWebBench (Liu et al., 2024c) and a 19.1% increase in element accuracy on Mind2Web (Deng et al., 2023). More surprisingly, we observed that this training generalizes to non-UI domains, resulting in improved performance in document understanding (Mathew et al., 2021), OCR (Singh et al., 2019; Liu et al., 2023c), and chart interpretation (Masry et al., 2022) tasks—outperforming even models specialized in these areas. These findings underscore the broader utility of web UI data as a powerful resource for improving text-rich visual understanding, enabling models to excel not only in UI tasks but across a diverse range of scenarios, including more complex agent tasks and non-UI domains.

## 2 DATASET CONSTRUCTION

In this section, we outline the process of constructing MultiUI. We developed an automated data collection pipeline by leveraging accessibility trees[1] and off-the-shelf LLMs. As illustrated in Figure 2, we construct the data collection pipeline through four stages: (1) raw website data scraping, (2) website curation, (3) task extraction from scraped websites, and (4) instruction construction.

### 2.1 WEBSITE RAW DATA SCRAPING

We begin by constructing a web raw dataset that includes HTML/CSS, high-resolution screenshots, and accessibility trees. The URLs in the "CC-MAIN-2024-10" dump from FineWeb (Penedo et al., 2024) are used to render websites via Playwright[2]. To prevent any potential contamination, we removed URLs that also appear in downstream benchmarks, such as VisualWebBench (Liu et al., 2024c). We employ the accessibility tree (refer to Appendix B for an example) as the text representation of a webpage for further utilization by LLMs because it typically provides a more compact structured representation compared to the raw HTML. The accessibility tree focuses on the most important visual elements, such as buttons, links, and headings, which are crucial for understanding the content and functionality of the page. It excludes non-essential elements like those related to styling or purely decorative purposes, which are often present in HTML but do not contribute to the

---

[1]https://developer.mozilla.org/en-US/docs/Glossary/Accessibility_tree
[2]https://github.com/microsoft/playwright

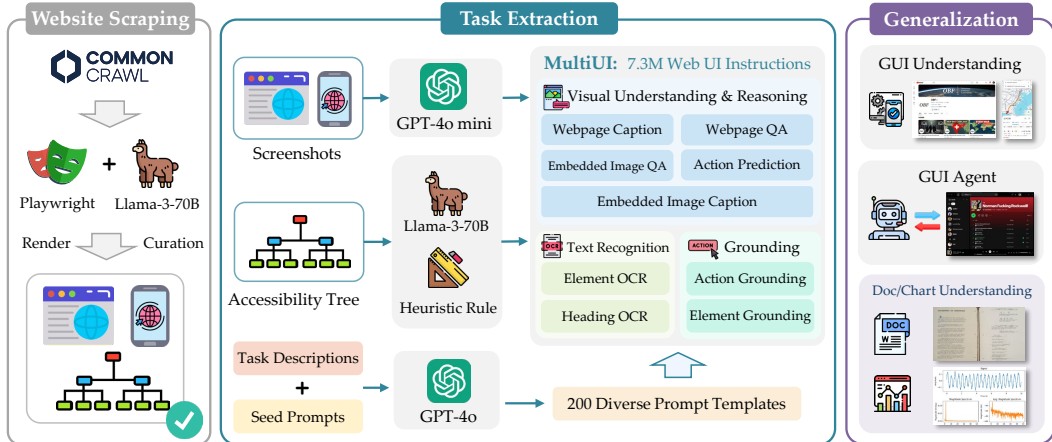

Figure 2: **Construction pipeline of MultiUI**. The process consists of four main stages: (1) Website Scraping; (2) Website Curation with Llama-3-70b-Instruct; (3) Task Extraction utilizing Llama-3-70b-Instruct, GPT-4o mini, and rule-based approaches to generate Web UI tasks across three categories: visual understanding and reasoning, text recognition, and grounding; (4) For each task, generate tasks samples by applying the diverse instruction templates paraphrased by GPT-4o.

core information. This higher information density in the accessibility tree makes it a more efficient and relevant input for language model processing, allowing the model to focus on key content without unnecessary noise. For details on raw data processing, see Appendix A. In total, 1.1 million websites were crawled. To ensure comprehensive coverage across diverse platforms and window size variations, we rendered websites on two simulated devices (Windows 10 and iPhone 12 Pro) using dynamic window size settings.

## 2.2 WEBSITE CURATION

Despite FineWeb's filtering, our crawled raw data still contains instances of inappropriate content and network errors. To address this issue, we employ an additional processing step using a filter language model. Specifically, we utilize the Llama-3-70B-Instruct (Dubey et al., 2024) as our filter model. It analyzes the accessibility tree of each website to identify problematic content, including adult material, gambling-related content, violence, discriminatory language, and network errors such as "403 Forbidden", "502 Bad Gateway", "Cloudflare blocking", "blank pages", and "404 Not Found" errors. Websites flagged as problematic are subsequently removed from the dataset. The prompt used by filter model can be found in Appendix C. After this step, approximately 10% of the raw websites were identified as harmful or invalid and subsequently removed, resulting in a raw dataset of 1 million website instances.

## 2.3 TASK EXTRACTION FROM SCRAPED WEBSITES

Existing approaches to using web content in multimodal models have limitations (Figure 1). Traditional rule-based methods for extracting images and surrounding text often introduce noise and lack contextual depth (Zhu et al., 2023; Schuhmann et al., 2021; 2022), while converting screenshots into simplified HTML structures imposes rigid formats (Lee et al., 2023; Gao et al., 2024), limiting generalization across domains. Additionally, methods that rely on GPT-4 to generate captions for web images (Chen et al., 2023) tend to overlook the rich interaction between text and visuals. Our approach aims to overcome these challenges by synthesizing general multimodal instructions from webpage UIs using text-based LLMs.

To provide multimodality models with robust perception, understanding, grounding, and reasoning capabilities, we construct a diverse set of tasks, featured by their different focuses of abilities to interact with the web. These types of tasks are crucial for effective web interaction, they are: (1) visual understanding and reasoning; (2) text recognition; (3) grounding. See Figure 3 for an overview of our constructed task samples. All prompts required in the sample construction process are meticulously refined through manual tuning, based on observations of the generated samples. Refer to

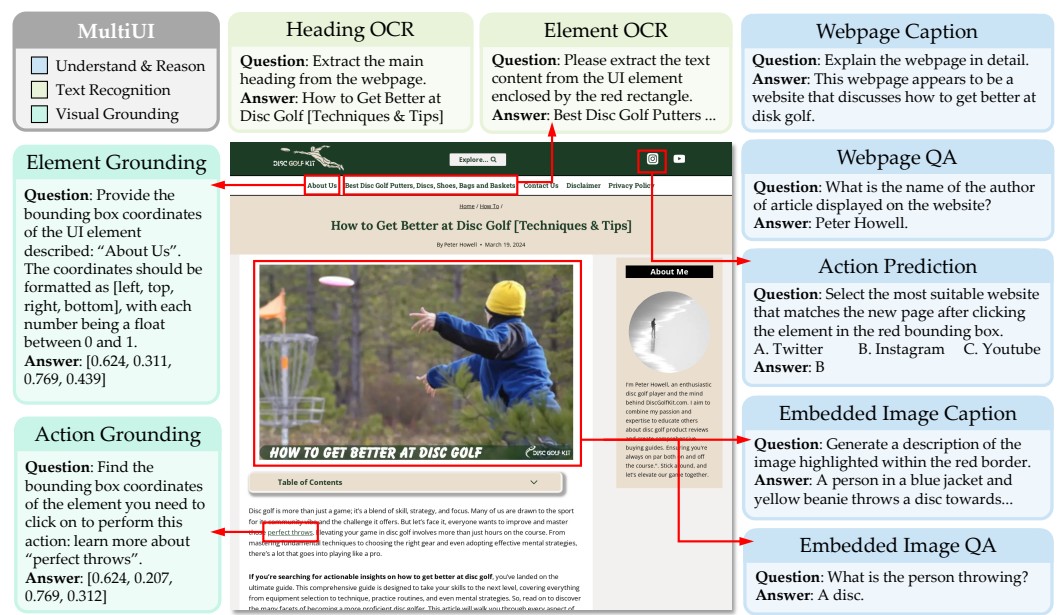

Figure 3: **Task samples from MultiUI.** To enhance multimodal models' perception, comprehension, grounding, and reasoning capabilities, we have designed a diverse set of nine tasks, emphasizing the critical abilities for text-rich visual understanding scenarios.

Appendix C for the complete set of prompts used for task extraction and Appendix D for examples of created task samples.

### 2.3.1 VISUAL UNDERSTANDING AND REASONING TASKS

Our visual understanding and reasoning tasks are designed to improve the model's ability to describe both the overall structure of web pages and the specific visual elements within them (Captioning), meanwhile, enhancing the models' abilities to answer questions (QA), and predict functionalities of elements (Action Prediction) about webpages.

**Webpage Captioning**: comprehending and summarizing the overall content and structure of a web page. The accessibility tree of each webpage serves as a concise textual representation that encodes the page's structural and semantic information. Therefore, we prompt Llama-3-70B-Instruct to synthesize the structured and informative accessibility tree into coherent and detailed descriptions.

**Webpage QA**: answering questions with respect to non-image content within webpages. Unlike Webpage Captioning, which provides a high-level summary of the entire page, this task targets detailed information extraction and reasoning about particular aspects of the page content. Given the accessibility tree, Llama-3-70B-Instruct is prompted to generate question-answer pairs. To improve the versatility and adaptability of this task, we design a dynamic in-context examples strategy and consider two kinds of styles of output answers. Specifically, a set of five questions from the LLaVA v1.5 instruction-tuning dataset are randomly sampled as in-context examples during each generation. This strategy helps the model draw from a broad range of patterns, reducing redundancy and increasing variability in question types. Furthermore, answers are provided in two distinct formats: detailed conversational responses to accommodate scenarios that require contextual richness, and concise, direct answers for use cases where brevity is preferred. The dual-format answers construction method ensures that the generated QA pairs are versatile and adaptable to different user needs.

**Embedded Image Captioning**: describing embedded images within a web page. While the accessibility tree provides a useful overview of the textual and interactive components of a webpage, it often falls short when dealing with embedded images. Descriptions for embedded images are often inadequate, with brief or even missing alt-text being a common issue. To address this issue, we employ the GPT-4o-mini[3] to generate rich and context-sensitive captions for these embedded im-

---

[3]https://openai.com/index/gpt-4o-mini-advancing-cost-efficient-intelligence/

| Platform | Visual Understanding and Reasoning | | | | | Grounding | | Text Recognition | | Total |
|---|---|---|---|---|---|---|---|---|---|---|
| | Web Capt. | Img Capt. | Web QA | Img QA | Act. Pred. | Action | Elem. | Head | Elem. | |
| Desktop | 150K | 526K | 1.1M | 979K | 65K | 1.2M | 694K | 98K | 175K | 5.0M |
| Mobile | 100K | 0 | 936K | 0 | 34K | 613K | 488K | 74K | 41K | 2.3M |
| Total | 250K | 526K | 2.1M | 979K | 99K | 1.8M | 1.2M | 172K | 217K | 7.3M |

Table 1: Statistics of our dataset MultiUI.

ages. The model is instructed to consider not only the visual features of the images but also their surrounding context provided by the accessibility tree, producing captions that are more informative and contextually relevant compared to standard alt-text tags.

**Embedded Image QA**: answering questions with respect to embedded images within webpages. Different from Embedded Image Captioning which provides general descriptions of images in the context of the webpage, Embedded Image QA focuses on answering specific questions about the visual content, requiring more detailed analysis. The Llama-3-70B-Instruct model is also employed to generate QA pairs from corresponding captions. The same dynamic in-context examples strategy and dual-format answers generation method as in Webpage QA are employed here.

**Action Prediction**: predicting the outcome of clicking a specific element on a webpage. Following Liu et al. (2024c), this task aims to predict the title of the redirected webpage after clicking a specified element, under a multiple-choice setting. Negative elements are randomly selected from the same webpage as the positive element and the titles of redirected sites are obtained through automated interaction using PlayWright.

### 2.3.2 OCR TASKS.

**Element OCR.** In this process, the HTML DOM tree of the crawled webpages is traversed to identify elements with textual descriptions exceeding 20 words, which are then utilized to create OCR task samples. Each task sample consists of a screenshot of the webpage, with a red bounding box highlighting the element targeted for OCR.

**Heading OCR.** Following Liu et al. (2024c), we incorporate Heading OCR to complement the Element OCR and provide a more comprehensive text recognition capability. It focuses on identifying and extracting the textual content of headlines or titles from web pages.

### 2.3.3 GROUNDING TASKS

**Action Grounding:** predicting the click position in response to a specific instruction, such as "learn more about perfect throws" as illustrated in Figure 3. Action grounding is crucial for developing web agents capable of performing actions autonomously based on user commands. Utilizing our processed accessibility data enhanced by the bounding boxes of elements, Llama-3-70b-Instruct is not only prompted to generate multiple grounding instructions but also provides the corresponding ground-truth bounding boxes.

**Element Grounding:** identifying the coordinates of an element based on its textual description. The training examples for this task are created by extracting textual descriptions alongside their corresponding bounding boxes from the HTML DOM tree.

For both grounding tasks, we implement two distinct settings: multi-choice and bounding box generation. In the multi-choice setting, eight candidate bounding boxes are presented within an input screenshot, and the model must select the correct one based on the given element description or instruction. In the bounding box generation setting, the model directly predicts the coordinates of the target element .

### 2.4 INSTRUCTION TEMPLATE CONSTRUCTION

We further diversify the instruction templates through a prompt variation approach (Appendix J).

We ultimately curated a dataset of 7.3 million web UI-related samples in the form of VQA, covering nine tasks across perception, comprehension, grounding, and reasoning capabilities, which we refer to as MultiUI. The statistics are shown in Table 1 and refer to Appendix D for examples.

## 3 EXPERIMENTAL SETUP

### 3.1 IMPLEMENTATION DETAILS

**Model Architecture**   We developed UIX using Qwen2-7B-Instruct (Yang et al., 2024) as the primary LLM backbone.   We also use Vicuna-7B-v1.5 (Chiang et al., 2023) and Llama-3.1-8B-Instruct (Meta, 2024) as backbones to further verify the effectiveness of our dataset. See Appendix E for details about our model architectures.

**Training Strategy**   To develop MLLMs with comprehensive GUI knowledge while maintaining robust general multimodal capabilities, we propose a two-stage training pipeline for our UIX models.

- **Stage 1: GUI Knowledge Learning.** In this stage, we fine-tune the model on 95% of MultiUI dataset to enhance its web/UI-related understanding capabilities.  This stage is crucial for developing the model's proficiency in interpreting and interacting with GUI environments.
- **Stage 2: Visual Instruction Tuning.** To help the model acquire robust general multimodal capabilities alongside enhanced GUI knowledge, we then continue the fine-tuning process using a combination of general visual instruction dataset (i.e., LLaVA–1.5 data[4] or LLaVA-NeXT data[5]) and the remaining 5% of the MultiUI data.

### 3.2 BENCHMARKS

We have selected diverse evaluation benchmarks, encompassing a variety of **in-domain GUI scenarios**, **out-of-domain OCR-related tasks**, **general multimodal tasks**, **agent benchmark**, to assess the models' capabilities in diverse multimodal tasks. See Appendix F for more details of these benchmarks.

- **GUI-Related Tasks.** We select VisualWebBench (Liu et al., 2024c), WebSRC (Chen et al., 2021), ScreenQA-Short (Hsiao et al., 2022), WidgetCap (Li et al., 2020) for GUI understanding. We also include Bbox-version of element and action grounding subtasks in VisualWebBench (Liu et al., 2024c), ScreenSpot (Cheng et al., 2024), RefExp (Wichers et al., 2018) for GUI grounding.
- **OCR-Related Tasks.**   DocVQA (Mathew et al., 2021), ChartQA (Masry et al., 2022), TextVQA (Singh et al., 2019), InfoVQA (Mathew et al., 2022), VisualMRC (Tanaka et al., 2021), OCRBench (Liu et al., 2023c).
- **General Multimodal Tasks.**   MMMU (Yue et al., 2024a), MMBench (Liu et al., 2023b) and VQA-V2 (Goyal et al., 2017). We also include RefCOCO+(REC) (Yu et al., 2016) to evaluate the grounding capability in natural image scenarios.
- **Agent Task.**   We employ Mind2Web (Deng et al., 2023) to train and evaluate the agent's capability of performing complex instructions on real-world websites.

We evaluate our models against various baselines, including LLaVA-1.5 series (Liu et al., 2024a) and LLaVA-1.6 (NeXT) series (Liu et al., 2024b).  To ensure a fair comparison, given the different backbones of UIX from the original LLaVA checkpoints, we re-implemented three baselines: LLaVA-Vicuna, Llama3.1, Qwen2, utilizing the same training data as LLaVA. We also compare our models with Pix2Struct (Lee et al., 2023), S4 (Gao et al., 2024), SeeClick (Cheng et al., 2024), CogAgent (Hong et al., 2023), and ScreenAI (Baechler et al., 2024). Furthermore, we include GPT-4V and GPT-4o as strong baselines to provide a comprehensive evaluation.

## 4 EXPERIMENTAL RESULTS AND ANALYSIS

### 4.1 RESULTS ON GUI-RELATED TASKS

**Significant Improvement on GUI Understanding and Grounding Benchmarks:** We present performance on GUI understanding and grounding benchmarks in Table 2.  Overall, our dataset

---

[4]https://huggingface.co/datasets/liuhaotian/LLaVA-Instruct-150K/blob/main/llava_v1_5_mix665k.json

[5]https://huggingface.co/datasets/lmms-lab/LLaVA-NeXT-Data

| Model | GUI Understanding | | | | GUI Grounding | | | |
|---|---|---|---|---|---|---|---|---|
| | Visual WebBench | Web SRC | SQA Short | Widget Cap | VWB Ele-G | VWB Act-G | SSpot | RefExp |
| GPT-4V (OpenAI, 2023) | 64.6 | - | - | - | 0.2 | 0 | 16.2 | - |
| GPT-4o (OpenAI, 2024) | - | - | - | - | - | - | 18.3 | - |
| Gemini 1.5 Pro (Reid et al., 2024) | 64.8 | - | - | - | - | - | - | - |
| Pix2Struct (Lee et al., 2023) | - | - | - | 136.7∗ | - | - | - | - |
| S4 (Gao et al., 2024) | - | 61.1∗ | - | 130.6∗ | - | - | - | - |
| SeeClick (Cheng et al., 2024) | 9.7 | - | - | - | - | - | **53.4** | - |
| CogAgent (Hong et al., 2023) | **28.7** | - | - | - | **29.3** | **36.6** | 47.4 | - |
| ScreenAI (Baechler et al., 2024) | - | **87.2**∗ | **94.8**∗ | **156.4**∗ | - | - | - | - |
| **Trained with LLaVA-1.5 data** | | | | | | | | |
| LLaVA-1.5-7B (Liu et al., 2023a) | 17.0 | 30.9 | 42.6 | 20.0 | 0.7 | 0.0 | 0.6 | 0.4 |
| LLaVA-1.5-13B (Liu et al., 2023a) | 19.4 | 32.5 | 46.0 | 10.2 | 0.0 | 0.0 | 0.9 | 1.1 |
| LLaVA-Vicuna† | 23.1 | 41.5 | 53.0 | 38.4 | 0.0 | 0.0 | 1.3 | 1.2 |
| Trained with LLaVA-1.5 data + MultiUI | | | | | | | | |
| UIX-Vicuna | **71.1** | **69.5** | **73.9** | **66.5** | **55.5** | **26.7** | **44.7** | **35.8** |
| Δ over LLaVA-Vicuna | +48.0 | +28.0 | +20.9 | +28.1 | +55.5 | +26.7 | +43.4 | +34.6 |
| **Trained with LLaVA-NeXT data** | | | | | | | | |
| LLaVA-NeXT-7B (Liu et al., 2023a) | 36.0 | 67.2 | 66.0 | 35.4 | 0.2 | 0.0 | 0.9 | 0.4 |
| LLaVA-NeXT-13B (Liu et al., 2023a) | 39.4 | 71.2 | 68.3 | 23.4 | 0.0 | 1.0 | 0.4 | 0.0 |
| LLaVA-NeXT-34B (Liu et al., 2023a) | 50.5 | **83.2** | 74.0 | 46.3 | 1.7 | 3.0 | 2.8 | 3.4 |
| LLaVA-NeXT-8B (Liu et al., 2024b) | 42.1 | 72.8 | 68.0 | 49.8 | 1.0 | 0.0 | 1.7 | 1.1 |
| LLaVA-Llama3.1† (Liu et al., 2024b) | 35.3 | 65.0 | 65.7 | 34.2 | 0.5 | 0.0 | 1.3 | 0.9 |
| LLaVA-Qwen2† (Liu et al., 2024b) | 41.7 | 72.5 | 68.6 | 38.0 | 1.2 | 0.0 | 1.3 | 1.9 |
| Trained with MultiUI+ LLaVA-NeXT data | | | | | | | | |
| UIX-Llama3.1 | 74.2 | 75.3 | 72.7 | 55.6 | 16.2 | 11.9 | 22.2 | 17.9 |
| Δ over LLaVA-Llama3.1 | +38.9 | +10.3 | +7.0 | +21.4 | +16.2 | +11.9 | +20.9 | +17.0 |
| UIX-Qwen2-7B | **75.9** | 82.9 | **78.8** | **72.7** | **66.1** | **35.6** | **55.2** | **43.5** |
| Δ over LLaVA-Qwen2 | +34.2 | +10.4 | +10.2 | +34.7 | +64.9 | +35.6 | +53.9 | +41.6 |

Table 2: **Results on GUI understanding and grounding benchmarks**. ∗ indicates specific fine-tuning on the corresponding training set. † denotes our re-implementation with the same backbone model architecture of UIX.

significantly improves model performances over all three backbones (i.e., LLaVA-Vicuna, LLaVA-Llama3.1, and LLaVA-Qwen2) as shown in light cyan rows. When training with LLaVA-1.5 data, the performance improvements on the nine benchmarks are all above 20%, up to 55.5%. The corresponding numbers training with LLaVA-1.6 data are 7.0% and 64.9%.

**Superior Performance over Larger Models:** In contrast, our UIX models, trained on the MultiUI dataset, demonstrate substantial performance improvements across GUI-related tasks. Despite having a relatively modest parameter count of 7B/8B, our models outperform baselines with significantly larger parameter counts, highlighting the efficacy of learning from text-rich web UI information. For instance, on VisualWebBench, UIX-Qwen2 achieves the highest score (75.9), surpassing all other MLLMs, including LLaVA-1.6-34B (50.5) and even GPT-4V (64.6).

**Enhanced GUI Grounding Capability:** Meanwhile, we observe that, compared with general figure grounding (Table 3), current MLLMs fail to handle grounding tasks in complex GUI scenarios. Conversely, trained with MultiUI, which encompasses two web grounding tasks, UIX also demonstrates enhanced GUI grounding capability.

## 4.2 RESULTS ON GUI AGENT TASKS

To further validate the effectiveness of the learned web UI knowledge, we evaluated UIX on Mind2Web (Deng et al., 2023), a web navigation GUI agent task.

Following previous work (Cheng et al., 2024), we use step-level success rate and element accuracy as the metrics. We include SeeClick (Cheng et al., 2024) and CogAgent (Hong et al., 2023) as baselines, both of which are trained on Mind2Web training set. See more experimental details in Appendix H.

| Model | General OCR / DocQA / ChartQA | | | | | | General Grounding |
|---|---|---|---|---|---|---|---|
| | Doc VQA | Chart QA | Text VQA | Info VQA | Visual MRC | OCR Bench | RefCOCO+ |
| GPT-4V (OpenAI, 2023) | 88.4 | 78.5 | 78 | 75.1 | - | 64.5 | - |
| GPT-4o (OpenAI, 2024) | 92.8 | 85.7 | 77.4 | 79.2 | - | 73.6 | - |
| Gemini 1.5 Pro (Reid et al., 2024) | 93.1 | 87.2 | 78.7 | 81.0 | - | - | - |
| Claude-3.5 Sonnet | 95.2 | 90.8 | 74.1 | 74.3 | - | - | - |
| Pix2Struct (Lee et al., 2023) | 76.6 | 58.6 | - | 40 | - | - | - |
| S4 (Gao et al., 2024) | - | 55.0 | - | - | - | - | - |
| CogAgent (Hong et al., 2023) | 81.6 | 68.4 | **76.1** | 44.5 | - | - | - |
| DocOwl-1.5-Chat (Hu et al., 2024a) | **82.2** | 70.2 | 68.6 | **50.7** | - | - | - |
| DocOwl2 (Hu et al., 2024b) | 80.7 | 70 | 66.7 | 46.4 | - | - | - |
| **Trained with LLaVA-1.5 data** | | | | | | | |
| LLaVA-1.5-7B (Liu et al., 2023a) | 28.1 | 18.1 | 46.0 | 25.8 | 35.3 | 31.3 | 50.0 |
| LLaVA-1.5-13B (Liu et al., 2023a) | 30.2 | 18.2 | 48.7 | 29.4 | 38.3 | 52.1 | 59.9 |
| LLaVA-Vicuna[†] | 46.1 | 21.2 | 59.6 | 31.9 | 39.7 | 38.1 | 61.7 |
| Trained with MultiUI + LLaVA-1.5 data | | | | | | | |
| UIX-Vicuna | **72.8** | **24.2** | **67.0** | **41.6** | **43.3** | **53.4** | **65.7** |
| Δ over LLaVA-Vicuna | +26.7 | +3.0 | +7.4 | +9.7 | +3.6 | +15.3 | +4.0 |
| **Trained with LLaVA-NeXT data** | | | | | | | |
| LLaVA-NeXT-7B (Liu et al., 2023a) | 74.4 | 54.8 | 64.8 | 37.0 | 33.3 | 52.1 | 77.0 |
| LLaVA-NeXT-13B (Liu et al., 2023a) | 77.5 | 62.4 | 67.0 | 41.5 | 35.9 | 55.0 | 80.8 |
| LLaVA-NeXT-34B (Liu et al., 2023a) | 83.9 | 68.6 | 69.4 | 51.3 | 37.9 | 57.2 | **84.8** |
| LLaVA-NeXT-8B (Liu et al., 2024b) | 78.2 | 69.2 | 65.3 | 37.6 | 29.3 | 55.2 | 79.5 |
| LLaVA-Llama3.1[†] | 74.7 | 66.5 | 64.3 | 35.7 | 46.8 | 54.0 | 74.8 |
| LLaVA-Qwen2[†] | 76.5 | 68.5 | 67.0 | 41.1 | 44.1 | 55.7 | 75.9 |
| Trained with MultiUI + LLaVA-NeXT data | | | | | | | |
| UIX-Llama3.1 | 78.0 | 66.9 | 65.1 | 44.2 | **49.7** | 58.6 | 71.7 |
| Δ over LLaVA-Llama3.1 | +3.3 | +0.4 | +0.8 | +8.5 | +2.9 | +4.6 | -3.1 |
| UIX-Qwen2 | **85.3** | **74.0** | **72.7** | **52.2** | 49.1 | **66.3** | 79.1 |
| Δ over LLaVA-Qwen2 | +8.8 | +5.5 | +5.7 | +11.1 | +5.0 | +10.6 | +3.2 |

Table 3: **Results on general OCR/Doc/Chart related QA and grounding benchmarks**. **Bold** text and underlined indicate the best-performing and the second-best models in each group, respectively.

| Model | Mind2Web | | | | | |
|---|---|---|---|---|---|---|
| | Cross-Task | | Cross-Website | | Cross-Domain | |
| | Step SR | Element Acc. | Step SR | Element Acc. | Step SR | Element Acc. |
| SeeClick (Cheng et al., 2024) | 25.5[†] | 28.3[†] | 16.4[†] | 21.4[†] | 20.8[†] | 23.2[†] |
| CogAgent (Hong et al., 2023) | 26.9 | 30.2 | 23.4 | 27.3 | 28.5 | 33.1 |
| LLaVA-Qwen2 | - | 7.5 | - | 7.6 | - | 10.4 |
| UIX-Qwen2 | - | 13.5 | - | 9.8 | - | 13.8 |
| LLaVA-Qwen2-M2W | 20.4 | 24.3 | 14.3 | 20.1 | 16.4 | 20.0 |
| UIX-Qwen2-M2W | **38.2** | **43.4** | **31.0** | **39.2** | **34.9** | **40.4** |
| Δ over LLaVA-Qwen2-M2W | +17.8 | +19.1 | +16.7 | +19.1 | +18.5 | +20.4 |

Table 4: Performance on a GUI agent task: Mind2Web. [†] indicates numbers taken from (Cheng et al., 2024). -M2W means further fine-tuning models on the Mind2Web training set.

**Superior Agent Performance compared with Larger Models:** As shown in Table 4, training on MultiUI improves our model's performance by up to 19.1% in element accuracy. After further training on the Mind2Web training set, UIX-Qwen2-M2W outperforms other existing GUI agent models significantly, surpassing both SeeClick and CogAgent in step success rate and element accuracy across all three test subsets, while UIX(7B) is smaller than SeeClick(9.6B) and CogAgent(18B) in model size.

## 4.3 RESULTS ON GENERAL DOCUMENT UNDERSTANDING AND GROUNDING TASKS

**Generalization to OCR/Doc/Chart Benchmarks:** As illustrated in Table 3, training on MultiUI yields substantial improvements in out-of-domain OCR-related figure comprehension. Notably,

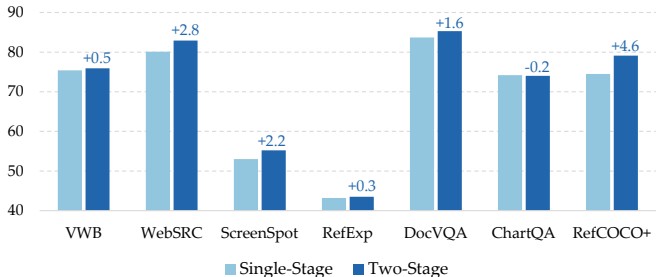

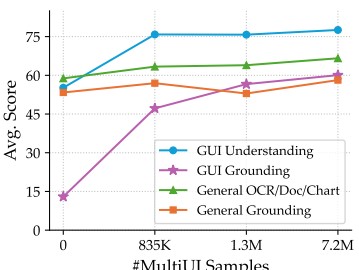

Figure 4: Comparison between single-stage and two-stage training strategies.

Figure 5: Effect of scaling up sample size.

UIX-Qwen2 outperforms both LLaVA-1.6-34B and DocOwl (Hu et al., 2024a;b), a model specifically designed for document understanding, across all evaluated OCR-related benchmarks. This success can be attributed to the diverse nature of MultiUI, which comprises an extensive collection of webpage screenshots, inherently covering a wide spectrum of OCR-related and abstract figure comprehension tasks.

**Transfer between Grounding Tasks:** Furthermore, the incorporation of web grounding instruction data during training demonstrates a significant enhancement in general grounding performance. UIX-Vicuna exhibits an improvement of 4.0% on RefCOCO+, when compared to models trained exclusively on general instruction data.

## 4.4 RESULTS ON GENERAL MULTIMODAL TASKS

**Robust General Multimodal Capabilities:** Here, we conduct an experiment to validate the efficacy of our MultiUI in maintaining general multimodal capabilities. As shown in Table 5, our UIX model trained with the mix of LLaVA data and MultiUI performs on par with the counterparts trained with only general visual instruction data. This demonstrates that our MultiUI dataset successfully enables models to acquire enhanced GUI knowledge alongside robust general multimodal capabilities simultaneously.

| Model | MMMU | MMBench | VQA-V2 |
|---|---|---|---|
| LLaVA-Llama3.1 | 38.8 | 72.0 | 80.0 |
| UIX-Llama3.1 | 42.3 | 74.7 | 80.0 |
| LLaVA-Qwen2 | 44.7 | 76.5 | 81.6 |
| UIX-Qwen2 | 41.8 | 77.4 | 82.1 |

Table 5: Performance on other general multimodal benchmarks.

## 4.5 ABLATIONS

**Two-stage training outperforms single-stage training** We conducted a comparative experiment to evaluate the effectiveness of the two-stage training mechanism (Section 4.5). As illustrated in Figure 4 the two-stage strategy outperforms the single-stage approach in both GUI and general scenarios. This result highlights the efficacy of the two-stage training strategy that achieves robust GUI knowledge while preserving strong general multimodal understanding capabilities.

**Model performance improves with more MultiUI data** To assess the scaling effect, we conducted an analysis by gradually increasing the volume of data in MultiUI and evaluating the aggregated performance across the four task categories. As illustrated in Figure 5, performance across all task categories generally improves as the data size increases. Specifically, GUI capabilities experience the most substantial improvements with increased data volume. GUI Grounding, in particular, demonstrates a remarkable increase, rising dramatically from approximately 12% to 60%.

**Different MultiUI tasks enhance unique abilities** To study the impact of different task types in MultiUI, we trained models with data in each individual task type and evaluated their performances. Figure 6 highlights several key findings from this study. In particular, both QA and Caption samples enhance performance in GUI understanding and OCR-related benchmarks, while OCR samples benefit most in GUI understanding. More detailed analysis can be found in Appendix I.

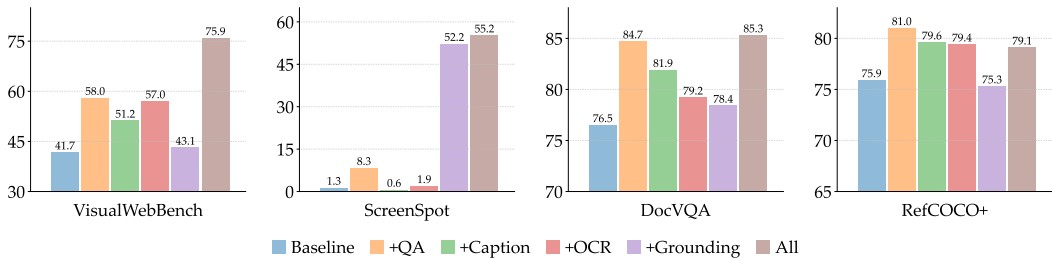

Figure 6: Ablation study of training on four task types.

# 5 RELATED WORK

**Web-Based Multimodal Pre-training Dataset Collection**    Web can be utilized as a valuable resource for collecting multimodal data for MLLM training. For example, the LAION series (Schuhmann et al., 2021; 2022) harvesting billions of image-text pairs from CommonCrawl[6] by extracting images with their corresponding alt-text tags from HTML. MMC4 (Zhu et al., 2023) and OBELICS (Laurençon et al., 2023) focus on an interleaved image-text format that can accommodate multiple images within a single instance. However, we argue that these datasets do not make full use of web UI information, since they cannot capture the rich layout information and diverse UI elements of webpages. In this paper, we propose that the accessibility tree serves as a well-structured and informative textual representation of a webpage, enabling powerful LLMs to be harnessed to generate diverse multimodal instructional data that encompasses perception, understanding, reasoning, and grounding capabilities.

Recently, there have also been some datasets focusing on pre-training on the rich structure information of webpage, such as Pix2Struct (Lee et al., 2023) which involves parsing screenshots into simplified HTML, and S4 (Gao et al., 2024) which focus collect multiple types training signals from the web. However, these methods either fall short in a specific task type so must be followed by further fine-tuning downstream tasks, or only focus on tasks in web scenarios and cannot easily generalize to domains beyond web pages and HTML.

**MLLMs for GUI-related Multimodal Tasks**    Graphical user interfaces (GUIs) serve as a pivotal medium for human-computer interactions. Recently, MLLMs have made significant strides in UI understanding, focusing on two key areas: information extraction from images and the development of foundation models for UI tasks. For information extraction, deep learning-based Optical Character Recognition (OCR) has advanced text reading capabilities (Baek et al., 2019). Concurrently, methods like Pix2Struct (Lee et al., 2023) and DocOwl (Hu et al., 2024a) have emerged for identifying and localizing UI elements within screenshots. Building on these element extraction techniques, several foundation models have demonstrated impressive UI comprehension abilities. Notable examples include ScreenAI (Baechler et al., 2024), CogAgent (Hong et al., 2023), and Ferret-UI (You et al., 2024). These models can perform tasks such as element detection, captioning, and interaction prediction using only screenshot inputs, without relying on additional metadata like DOM structures. Concurrently, UGround (Gou et al., 2024) proposes a strong universal visual grounding model for GUI agents. Different from these existing works, we focus on leveraging structured textual information from web UIs to enhance MLLMs' text-rich visual understanding capabilities across diverse domains. Our approach utilizes text-based LLMs to synthesize general multimodal instructions from webpage accessibility trees, creating a large-scale dataset that improves model performance not only on web UI tasks but also generalizes well to non-UI domains.

# 6 CONCLUSION

In this work, we tackled the challenge of improving text-rich visual understanding in MLLMs by leveraging webpage UIs as a naturally structured and diverse data source. We introduced the dataset MultiUI, which we demonstrated to significantly improves model across domains. Our experiments also underscore the importance of structured web data as a training resource, which helps MLLMs process and interact with text-rich visual environments more effectively.

---

[6]https://commoncrawl.org/

ACKNOWLEDGEMENT

Junpeng Liu and Wai Lam are supported by a grant from the Research Grant Council of the Hong Kong Special Administrative Region, China (Project Code: 14200620).

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

# Table of Contents in Appendix

## A    SCREENSHOT CROPPING & ACCESSIBILITY TREE REFINEMENT

The original full-page webpage screenshots may be quite long (e.g., height is much larger than width), to produce screenshots that are suitable to be input into an image-based model with image size or resolution requirements, we crop the full-page screenshots. Because webpage screenshots are typically structured vertically, horizontal cuts preserve the logical flow of text and sections. Thus, we crop the screenshots horizontally by selecting a random height while keeping the width fixed as the original one. Specifically, a height-width ratio is sampled from a uniform distribution [min_height_width_ratio, max_height_width_ratio], where the ranges are [0.5, 1.5] and [1.5, 2.5] for Windows 10 and iPhone 12 Pro, respectively. These ranges are determined by considering screenshot sizes in existing GUI datasets (e.g., VisualWebBench, ScreenSpot).

Required by grounding tasks samples, which need element coordinates, we extend the accessibility tree as used in Zhou et al. (2024) by adding the bounding boxes of each element into the original accessibility tree. Figure 7 illustrates an example of the processed accessibility tree of a webpage. The accessibility tree is comprised of several lines, each detailing a UI element. Each schema includes: 1) The UI element ID, 2) The UI element type, 3) the text content (e.g., OCR text, icon name), and 4) The bounding box coordinates.

# B  EXAMPLE OF OUR PROCESSED ACCESSIBILITY TREE

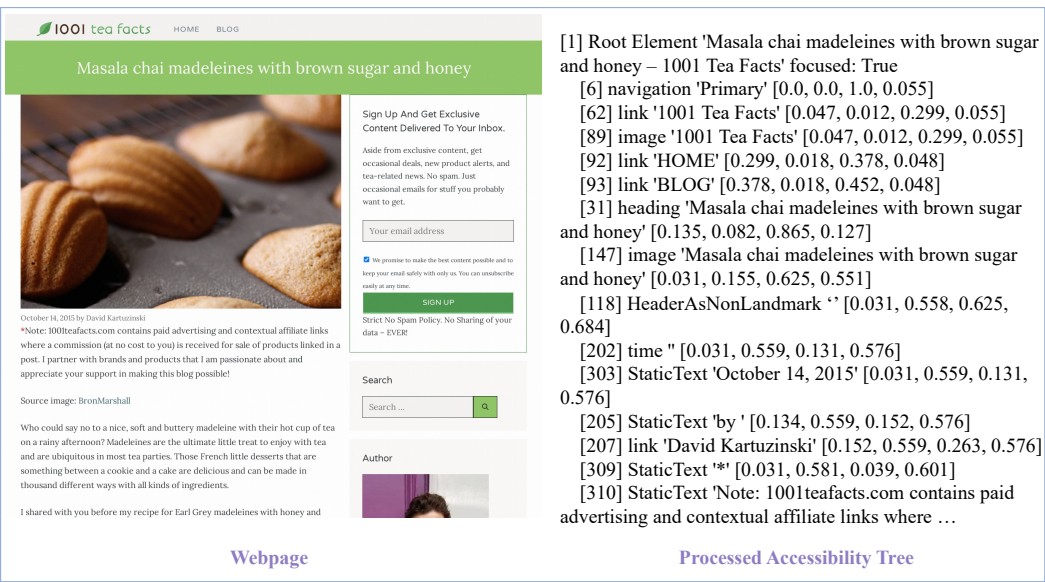

Figure 7: **Example of processed accessibility tree**. (Left) The webpage screenshot; (Right) The corresponding accessibility tree. Each line represents an element in the webpage, formatted as <element-id> <element-type> <embedded text> <bounding-box coordinates>

## C    PROMPTS FOR TASK EXTRACTION

**Prompt for Website Curation.**

You are given a truncated accessibility tree of a webpage, which is derived by a crawling program:

{axtree}

You need to answer the following two questions: whether the crawl is successful (e.g., 403 forbidden, 502 bad gateway, blocked by Cloudflare, blank page, 404 not found are considered as unsuccessful etc. )? Does the webpage contain some harmful content (e.g., adult content, gambling, violence, discrimination)

Your output only consists of two lines, each line is your final answer to one question. Only output "YES" or "NO" each line and do not generate any other content. Note that the truncation does not necessarily mean the crawl is not successful.

**Prompt for Constructing Webpage Captioning Samples.**

You are an AI visual assistant that can analyze a single screenshot of a webpage. You receive the meta description extracted from its HTML content and the accessibility tree of the webpage.

The accessibility tree consists of several lines, each describing an UI element. Each schema contains: 1) The UI element IDs, 2) The UI element types, 3) The OCR text (when applicable) or the element descriptions (e.g. captioning, or the icon name), and 4) The bounding box coordinates [left, top, right, bottom], as [x1, y1, x2, y2] with floating numbers ranging from 0 to 1. These values correspond to the left, top, right, and bottom. Indentation are used to indicate the hierarchical structure between the elements.

Using the provided meta description and accessibility tree, describe the webpage in a detailed manner.

Instead of directly mentioning the bounding box coordinates, utilize this data to explain the webpage using natural language. Include details like element counts of a specific UI element type, position of the UI elements, relative position between the UI elements.

Important Notes about your caption:
(1) When using the information from the meta description and the accessibility tree, directly explain the webpage, and do not mention that the information source is the meta description or the accessibility tree (e.g., "Based on the provided meta description and accessibility tree, I can describe the webpage as follows").
(2) When you mention some text or paragraph, mention the summarized content instead of just saying "there are some news articles or text".
(3) Every time when you want to express "below", "right", "left", "following", "above" and other words representing relative position, you must carefully compare the bounding coordinates of elements you want to mention, because the orders within the accessibility tree do not mean the relative position, e,g, element "A" appearing after another element "B" within the accessibility tree may be on top of the other element "B", in term of bounding box coordinates.
(3) Every time when you want to express "top", "right", "left", "bottom", "above" and other words representing absolute position, you must determine its position based on the reference points: top left [0, 0, 0, 0], top right [1.0, 0, 1.0, 0], bottom left [0, 1.0, 0, 1.0], and bottom right [1.0, 1.0, 1.0, 1.0].
(4) No need to discuss overall design, e.g., "The overall layout is clean and organized, with clear headings and concise text").
(5) You cannot say there are no images on the page.
(6) Do not mention line breaks and separator line.

**Prompt for constructing Webpage QA Samples.**

You are given the following webpage, describe in words. Follow the following guidance to think step by step before generating five questions and their answers.

(Webpage Mete Description) The meta description of the current web page is "description". Based on both the web page description and your understanding, think carefully what the website can be used to do.

(Webpage Accessibility Tree) You are given the following accessibility tree, Each schema contains: 1) The UI element IDs, 2) The UI element types, 3) The OCR text (when applicable) or The element descriptions (e.g. captioning, or the icon name), 4) The bounding box coordinates [left, top, right, bottom], as [x1, y1, x2, y2] with floating numbers ranging from 0 to 1. These values correspond to the left, top, right, and bottom. Indentation are used to indicate the hierarchical structure between the elements.
Note: The appearing orders of elements in the accessibility tree do not represent their relative vertical positions. For instance, even if element "A" comes after element "B" in the accessibility tree, it might still appear above element "B" based on their bounding box coordinates (y1, y2 of "A" values are smaller than those of "B", so "A" is above "B" vertically). You must determine their vertical positions by carefully comparing their y1, y2 coordinates instead of infering their positions based on the orders within the accessibility tree. axtree

(Potential Questions) Some example questions on other websites are shown below for your reference. {question_demo}

(Answers) For each question, you should generate two kinds of answers: short answer and detailed answer. Ask questions whose short answers consist of fewer than 10 words, and the answer should be as short as possible.
The detailed answer consists of thinking or reasoning process of obtaining the final answer and should be as detailed as possible. (Notes)
1) Your questions should be as hard as possible, and need deep understanding about the webpage and reasoning ability.
2) You must ask objective questions, not subjective ones.
3) Your questions should be as diverse as possible.
4) Do not mention in answers that you answer questions based on accessbility tree, instead, answer as if you are seeing the webpage screenshot.
5) The given meta description is only for your better understanding about the webpage, so you should never ask any questions specific to the meta description.
6) You can only generate questions that can be answered immediately based on the current webpage, do not ask those that need your background knowledge.
7) You should never ask questions about the advertisements or the whether-accept-cookie-or-not section of the webpage, if any.
8) Do not user element IDs to refer an element.
9) Do not ask questions that you cannot answer based on the given webpage accessibility tree.
10) Do not ask any questions about the main heading of the webpage, like "what is the title of the webpage? ", "What is the title of the main heading? ".

(Final Output) Your output should be several lines of json with each line being an question-answer pair, and do not generate any other content. You only speak JSON. Do not write text that isn't JSON.

{{"question": <the question content, string data type enclosed in double quotes>, "answer": <short answer, string data type enclosed in double quotes>, "detailed_answer": <detailed answer, string data type enclosed in double quotes>}}

**Prompt for constructing Embedded Image QA Samples.**

(Caption)
[The start of the caption]
{caption}
[The end of the caption]

(Potential Questions)
Some example questions on other websites are shown below for your reference.
{question_demo}

(Answers)
For each question, you should generate two kinds of answers: short answer and detailed answer. Ask questions whose short answers consist of fewer than 10 words, and the answer should be as short as possible. The detailed answer consists of thinking or reasoning process of obtaining the final answer and should be as detailed as possible.

(Notes)
1) Your questions should be as hard as possible, and need deep understanding about the webpage and reasoning ability.
2) You must ask objective questions, not subjective ones. 3) Your questions should be as diverse as possible.
4) Do not mention in answers that you answer questions based on the caption, instead, answer as if you are seeing the webpage screenshot.
5) You can only generate questions that can be answered immediately based on the current webpage, do not ask those that need your background knowledge.
6) You should never ask questions about the advertisements or the whether-accept-cookie-or-not section of the webpage, if any.
7) Do not ask questions that you cannot answer based on the caption.
8) Do not ask any questions about the main heading of the webpage, like "what is the title of the webpage? ", "What is the title of the main heading? ".

(Final Output)
Your output should be several lines of json with each line being an question-answer pair, and do not generate any other content. You only speak JSON. Do not write text that isn't JSON.
{{"question": <the question content, string data type enclosed in double quotes>, "answer": <short answer, string data type enclosed in double quotes>, "detailed_answer": <detailed answer, string data type enclosed in double quotes>}}

**Prompt for constructing Action Grounding Samples.**

The screenshot below shows the webpage you see. Follow the following guidance to think step by step before generating several executable instructions on it.

(Webpage Mete Description)
The meta description of the current web page is "{description}". Based on both the web page description and your understanding, think carefully what the website can be used to do.

(Webpage Accessibility Tree)
You are given the following accessibility tree, Each schema contains: 1) The UI element IDs, 2) The UI element types, 3) The OCR text (when applicable) or The element descriptions (e.g. captioning, or the icon name), 4) The bounding box coordinates (left, top, right, bottom), quantized and normalized between 0 and 999.
Indentation are used to indicate the hierarchical structure between the elements.
{axtree}

(Potential Instructions and Target Bounding Boxes)
Think what instructions would be taked by human to the above interactive elements on the current webpage. Describe the instruction (i.e., action description) using natural expression, and output the corresponding bounding box (exactly same as in the accessibility tree). Some example instructions on other websites are shown below.
1) search in the forum
2) go to personal homepage
3) create a new issus
4) view orders
5) zoom in the map
6) check my postbox
Notes:
1) The instructions above are only for reference and the generated instructions must be executable on the current web page and as diverse as possible.
2) Do not generate action description about accepting or decline cookie policy
3) Because the generated action descriptions are used to test the capability of other multi-modal models, your action description should not be too detailed and simple, even include the detailed position of the target element. E.g., click the red log-in button at bottom-right side
4) Your generated action description / instruction must only correspond to one element within the screenshot, in order not to lead to ambiguous answers.
5) Your generated action description / instruction must corresponds to a interactive element within the screenshot, e.g., button, input field, link, etc, instead of those not interactive, e.g., StaticText.
6) Do not user element IDs to refer an element.
7) Do not generate instructions targeted on the root element named "Root Element".

(Final Answer)
Finally, conclude your answer with the format below strictly, comprising one to five single-step executable instructions / action descriptions on the current webpage. Note that you can output a line whose content is "Invalid]" if you find no proper instructions can be generated and you don't have to force yourself to generate one. Your answer should be several lines of json or a line with "[Invalid]", and do not generate any other content. You only speak JSON. Do not write text that isn't JSON.
Output format:
{{"instruction": <the action description / instruction, string data type enclosed in double quotes>, "bbox": <the corresponding bounding box coordinate copy in the accessibility tree, list of integers>}}

## D EXAMPLE TRAINING DATA BY TASK

### Webpage Captioning

**Prompt:** Describe all the visual and textual components of the webpage comprehensively.

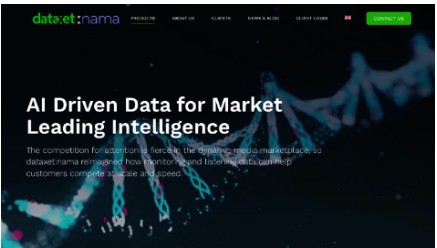

**Response:** The webpage appears to be a product or company page, with a prominent logo displayed at the top left corner, repeated five times. Below the logo, there is a navigation menu with seven links: "PRODUCTS", "ABOUT US", "CLIENTS", "NEWS & BLOG", "CLIENT LOGIN", and "English", which has a small flag icon next to it. These links are positioned horizontally, taking up most of the top section of the page. In the middle of the page, there is a large heading that reads "AI Driven Data for Market Leading Intelligence", followed by a subheading that explains how the company, dataxet:nama, helps customers compete in the media marketplace by reimagining monitoring and listening data. This text is centered on the page and takes up a significant amount of vertical space. There is no prominent image or graphic on the page, aside from the logo and the small flag icon next to the "English" link. The overall layout is focused on presenting the company's message and navigation options.

Figure 8: Example of training data for webpage captioning.

### Embedded Image Captioning

**Prompt:** Give an in-depth description of the picture inside the red marked area.

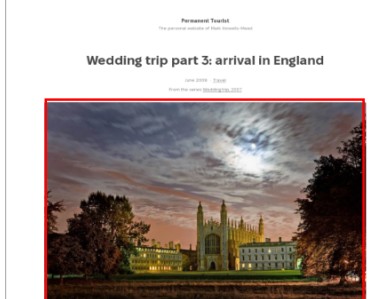

**Response:** The image depicts a stunning, gothic-style chapel, likely King's College Chapel at the University of Cambridge, illuminated during the night. The chapel stands tall with its intricate stone architecture, large pointed windows, and spires reaching towards a sky filled with textured clouds, dramatically lit by the moon. The moonlight and artificial lighting create a serene yet slightly eerie ambiance. The foreground is dominated by a grassy field and trees, which frame the structure and add to the overall picturesque and timeless feel of the scene.

Figure 9: Example of training data for embedded image captioning.

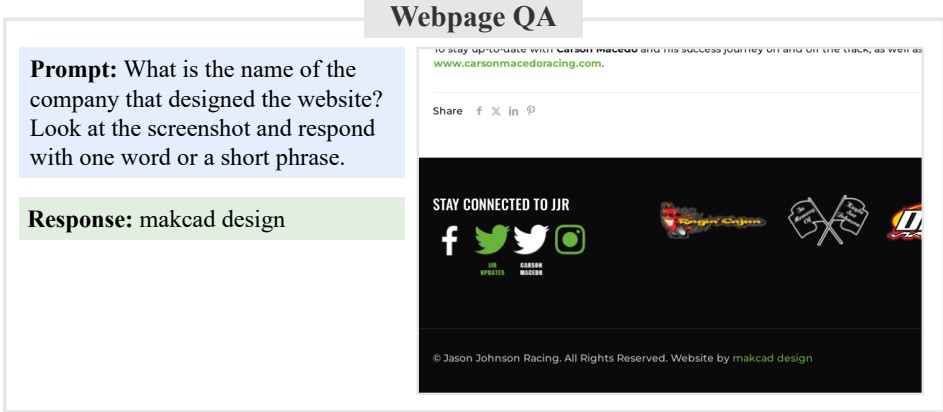

Figure 10: Example of training data for webpage QA.

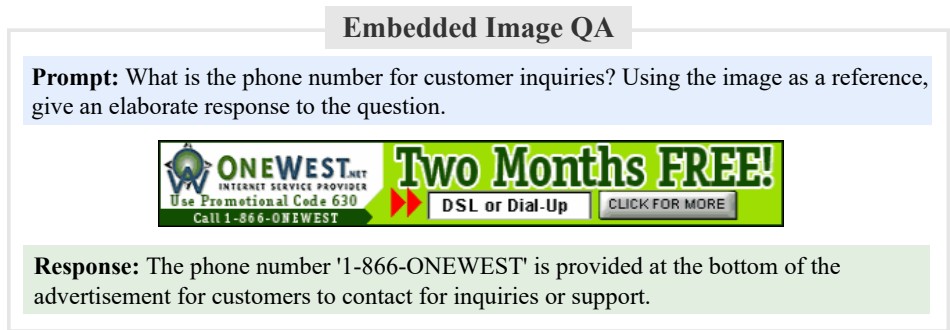

Figure 11: Example of training data for embedded image QA.

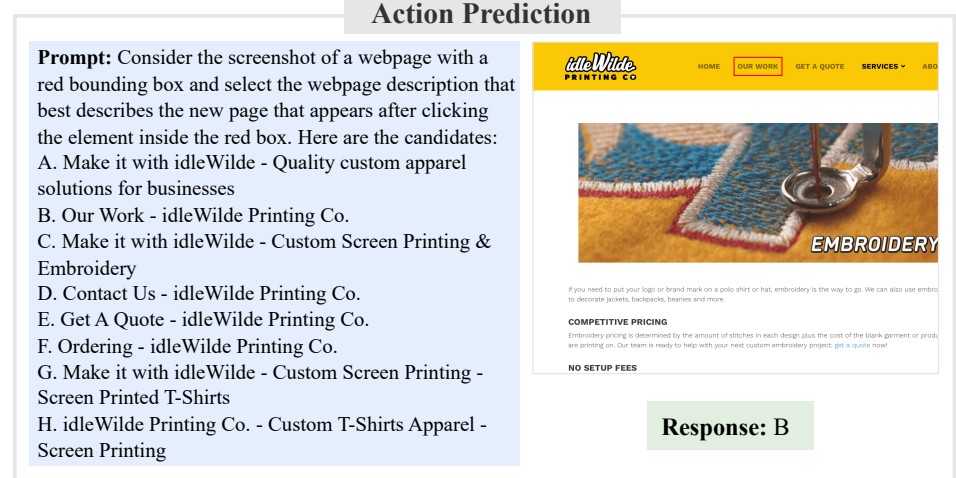

Figure 12: Example of training data for action prediction.

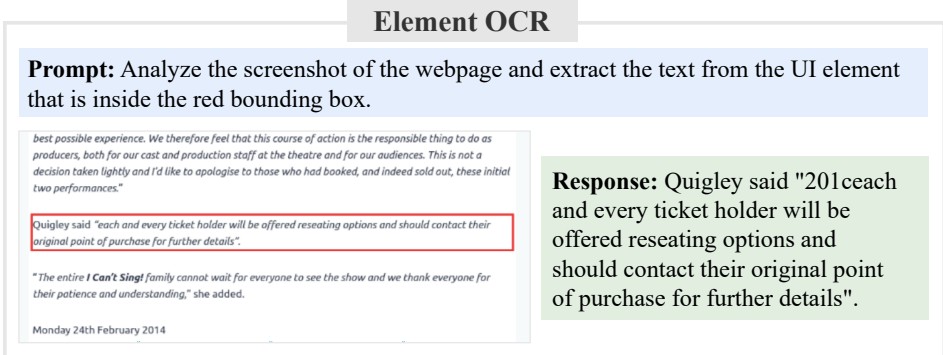

Figure 13: Example of training data for element OCR.

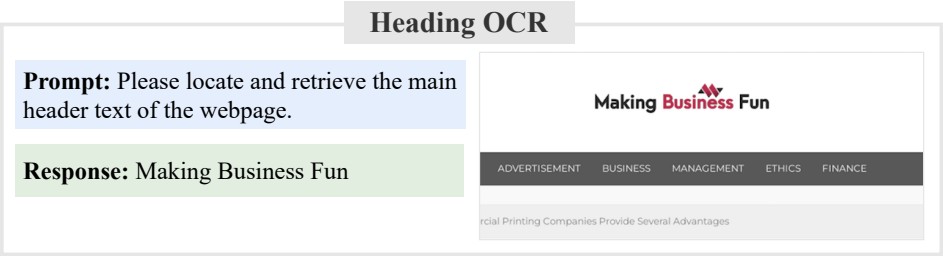

Figure 14: Example of training data for heading OCR.

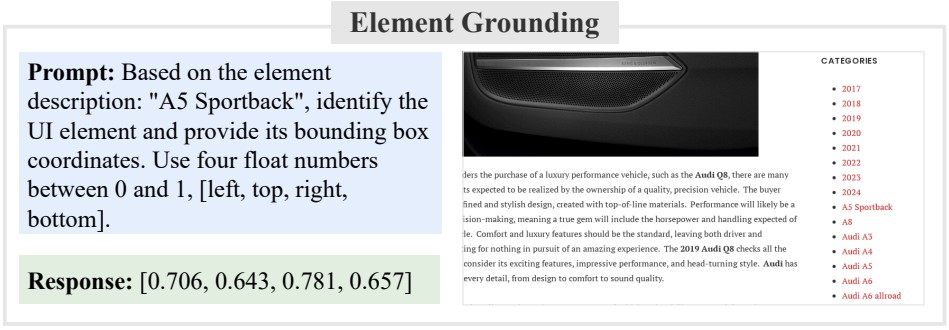

Figure 15: Example of training data for element grounding.

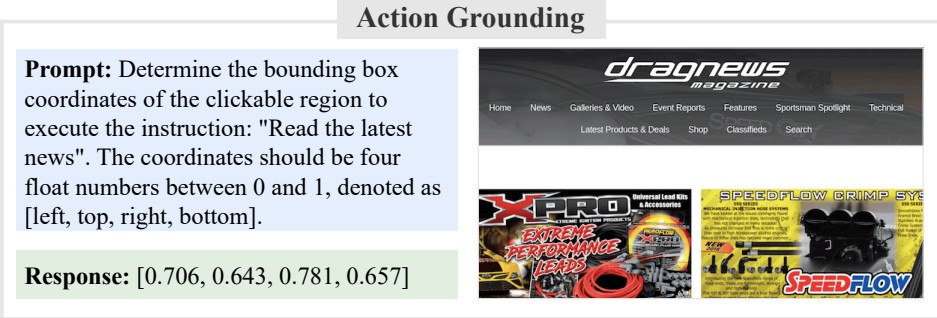

Figure 16: Example of training data for action grounding.

# E    TRAINING SETUP

To leverage the high-resolution images collected in Section 2, we adopted a dynamic high-resolution strategy inspired by LLaVA-NeXT (Liu et al., 2024b). The process involves dividing the input image into patches matching the vision encoder's original resolution, encoding these patches independently, and combining them into a single feature map. We also incorporate a downsampled version of the entire image to provide global context. This combined representation is then processed by the LLM, which allows us to scale the input to arbitrary resolutions while maintaining data efficiency.

| Model | LLM | Vision Encoder | Max Res. | Training Data |
|---|---|---|---|---|
| UIX-Vicuna | Vicuna-7B-v1.5 | CLIP | $672 \times 672$ | LLaVA 1.5 + MultiUI |
| UIX-Llama3.1 | Llama-3.1-8B-Instruct | CLIP | $672 \times 672$ | LLaVA NeXT + MultiUI |
| UIX-Qwen2 | Qwen2-7B-Instruct | Siglip | $768 \times 768$ | LLaVA NeXT + MultiUI |

Table 6: Details of three models developed in this paper.

## F DETAILS OF EVALUATED BENCHMARKS

**GUI Understanding Benchmarks**

- **VisualWebBench** (Liu et al., 2024c) is a multimodal benchmark specifically designed to evaluate the performance of Multimodal Large Language Models (MLLMs) in web-related tasks. Unlike existing benchmarks, it focuses on capturing the unique aspects of web pages and measures fine-grained abilities such as OCR, understanding, and grounding, providing a comprehensive evaluation of MLLMs in the web domain.

- **WebSRC** (Chen et al., 2021) is designed to evaluate models on structural reading comprehension tasks involving web pages, where both text understanding and structural comprehension are required.

- **ScreenQA** (Hsiao et al., 2022) bridges the gap between component-level and high-level task understanding in mobile app screen content. By annotating 86K question-answer pairs over the RICO dataset, this benchmark offers a comprehensive evaluation of screen reading comprehension in various application scenarios.

- **WidgetCap** (Li et al., 2020) evaluates the ability to generate natural language descriptions for mobile UI elements, a task critical for accessibility and enhancing language-based interaction.

**GUI Grounding**

- **ScreenSpot** (Cheng et al., 2024) serves as a comprehensive GUI grounding benchmark, covering mobile, desktop, and web environments. By encompassing multiple device types, it enables a thorough assessment of cross-platform GUI grounding capabilities.

- **RefExp** (Bai et al., 2021) is designed to assess the model's ability to predict the UI component referred to by a natural language expression, given an app screenshot. In order to simulate a more realistic scenario, we evaluate models under the setting of predicting coordinates directly instead of retrieving a component from a set of candidate components.

**GUI Agent Benchmark**

- **Mind2Web** (Deng et al., 2023) is the first dataset for training and evaluating generalist web agents to perform complex tasks on real-world websites. It includes over 2,000 tasks from 137 websites across 31 domains, offering diverse interactions for building robust web agents.

**General OCR / DocQA / ChartQA Benchmarks**

- **DocVQA** (Mathew et al., 2021) is a benchmark for evaluating models' ability to answer questions based on document images. This dataset challenges models to comprehend and extract relevant information from diverse document formats such as letters, forms, and reports.

- **ChartVQA** (Masry et al., 2022) presents a unique evaluation scenario where models are tested on their ability to answer complex reasoning questions about data visualized in charts. This benchmark is crucial for evaluating a model's capability to perform logical and arithmetic operations, as well as to reference visual features within the charts.

- **TextVQA** (Singh et al., 2019) focuses on questions that require reading and understanding text within images. It evaluates a model's ability to handle text-based reasoning, which is essential for tasks involving visually impaired users and real-world scenarios.

- **InfoVQA** (Mathew et al., 2022) focuses on the automatic understanding of infographic images through Visual Question Answering (VQA). This benchmark requires models to integrate reasoning over textual content, graphical elements, and data visualizations, which emphasizes both elementary reasoning and arithmetic skills.

- **VisualMRC** (Tanaka et al., 2021) focuses on evaluating the machine's ability to comprehend the visual layout and textual content within document images to answer questions. It requires a model to be able to read and reason about multiple pieces of text and non-text data in images and to generate abstractive answers.

- **OCRBench** (Liu et al., 2023c) encompasses 29 datasets and enables a thorough assessment across a wide range of text-related visual tasks, such as Text Recognition, Scene Text-Centric Visual Question Answering, Document-Oriented VQA, Key Information Extraction, and Handwritten Mathematical Expression Recognition.

**General-domain Tasks**

- **MMMU** (Yue et al., 2024a) is a comprehensive benchmark designed to assess multimodal models on complex, college-level tasks across six core disciplines, including Art & Design, Business, Science, Health & Medicine, Humanities & Social Science, and Tech & Engineering.

- **MMBench** (Liu et al., 2023b) is a bilingual benchmark for evaluating the multimodal capabilities of large vision-language models (VLMs). It features a diverse set of carefully crafted evaluation questions and employs rigorous quality control measures, enabling precise assessments of model performance in both English and Chinese contexts.

- **VQAv2** (Goyal et al., 2017) introduces a balanced dataset that pairs each question with two similar images leading to different answers. The findings show that state-of-the-art VQA models struggle with this dataset, underscoring the need for models to better utilize visual information rather than relying on language priors.

- **RefCOCO+** (Yu et al., 2016) focuses on generating and comprehending natural language referring expressions for objects in images, particularly by improving the use of visual context.

# G EVALUATION DETAILS AND FULL RESULTS

In this section, we describe the evaluation details and present the full experimental results of both baselines and our UIX models.

We employ LMMs-Eval (Zhang et al., 2024) for all evaluations except for VisualWebBench, for which we use official evaluation code[7]. The 7 shows the employed evaluation metrics for benchmarks. Table 8 and Table 9 show results of baselines and our UIX models.

| Benchmarks | Metric | Split |
|---|---|---|
| VisualWebBench | Aggregated Score | test |
| WebSRC | SQuAD-F1 | validation |
| ScreenQA-short | SQuAD-F1 | test |
| WidgetCap | CIDEr | test |
| Element Ground (VWB) | Accuracy (IoU>0.5) | test |
| Action Ground (VWB) | Accuracy (IoU>0.5) | test |
| ScreenSpot | Accuracy (IoU>0.5) | test |
| RefExp | Accuracy (IoU>0.5) | test |
| DocVQA | ANLS | validation |
| ChartQA | Relaxed Accuracy | test |
| TextVQA | Exact Match | validation |
| InfoVQA | ANLS | validation |
| VisualMRC | ROUGE-L | test |
| OCRBench | Accuracy (%) | test |
| RefCOCO+ (REC) | Accuracy (IoU>0.5) | validation |

Table 7: Evaluation metrics for all benchmarks.

| | LLaVA-1.5-7B | LLaVA-1.5-13B | LLaVA-1.6-7B | LLaVA-1.6-13B | LLaVA-1.6-34B | LLaVA-NeXT-8B |
|---|---|---|---|---|---|---|
| VisualWebBench | 17.0 | 19.4 | 36.0 | 39.4 | 50.5 | 42.1 |
| WebSRC | 30.9 | 32.5 | 67.2 | 71.2 | 83.2 | 72.8 |
| ScreenQA-short | 42.6 | 46 | 66 | 68.3 | 74 | 68 |
| WidgetCap | 20 | 10.2 | 35.4 | 23.4 | 46.3 | 49.8 |
| Element Ground (VWB) | 0.73 | 0 | 0.24 | 0 | 1.7 | 0.97 |
| Action Ground (VWB) | 0 | 0 | 0 | 0.99 | 3 | 0 |
| ScreenSpot | 0.6 | 0.9 | 0.9 | 0.4 | 2.8 | 1.7 |
| RefExp | 0.4 | 1.1 | 0.4 | 0 | 3.4 | 1.1 |
| DocVQA | 28.1 | 30.2 | 74.4 | 77.5 | 83.9 | 78.2 |
| ChartQA | 18.1 | 18.2 | 54.8 | 62.4 | 68.6 | 69.2 |
| TextVQA | 46 | 48.7 | 64.8 | 67 | 69.4 | 65.3 |
| InfoVQA | 25.8 | 29.4 | 37 | 41.5 | 51.3 | 37.6 |
| VisualMRC | 35.3 | 38.3 | 33.3 | 35.9 | 37.9 | 29.3 |
| OCRBench | 31.3 | 33.6 | 52.1 | 55 | 57.2 | 55.2 |
| RefCOCO+ | 50 | 59.9 | 77 | 80.8 | 84.8 | 79.5 |
| MMMU | 36.3 | 35.4 | 36.3 | 35 | 49.3 | 40.3 |
| MMBench | 64.2 | 68.5 | 67.1 | 69.2 | 78.1 | 72.2 |
| VQAv2 | 76.1 | 77.8 | 79.9 | 80.6 | 81.8 | 80.7 |

Table 8: Full experimental results of baselines.

---

[7]https://github.com/VisualWebBench/VisualWebBench

| | LLaVA-Vicuna | UIX-Vicuna | LLaVA-Llama3.1 | UIX-Llama3.1 | LLaVA-Qwen2 | UIX-Qwen2 |
|---|---|---|---|---|---|---|
| VisualWebBench | 23.1 | 71.1 | 35.3 | 74.2 | 41.7 | 75.9 |
| WebSRC | 41.5 | 69.5 | 65.0 | 75.3 | 72.5 | 82.9 |
| ScreenQA-short | 53.0 | 73.9 | 65.7 | 72.7 | 68.6 | 78.8 |
| WidgetCap | 38.4 | 66.5 | 34.2 | 55.6 | 38.0 | 72.7 |
| Element Ground (VWB) | 0.0 | 55.5 | 0.5 | 16.7 | 1.2 | 66.1 |
| Action Ground (VWB) | 0.0 | 26.7 | 0.0 | 11.9 | 0.0 | 35.6 |
| ScreenSpot | 1.3 | 44.7 | 1.3 | 22.2 | 1.3 | 55.2 |
| RefExp | 1.2 | 35.8 | 0.9 | 17.9 | 1.9 | 43.5 |
| DocVQA | 46.1 | 72.8 | 74.7 | 78.0 | 76.5 | 85.3 |
| ChartQA | 21.2 | 24.2 | 66.5 | 66.9 | 68.5 | 74.0 |
| TextVQA | 59.6 | 67.0 | 64.3 | 65.1 | 67.0 | 72.7 |
| InfoVQA | 31.9 | 41.6 | 35.7 | 44.2 | 41.1 | 52.2 |
| VisualMRC | 39.7 | 43.3 | 46.8 | 49.7 | 44.1 | 49.1 |
| OCRBench | 38.1 | 53.4 | 54.0 | 58.6 | 55.7 | 66.3 |
| RefCOCO+ | 61.7 | 65.7 | 74.8 | 71.7 | 75.9 | 79.1 |
| MMMU | 34.7 | 33.6 | 38.8 | 42.3 | 44.7 | 41.8 |
| MMBench | 66.1 | 66.9 | 72.0 | 74.7 | 76.5 | 77.4 |
| VQAv2 | 78.5 | 79.8 | 80.0 | 80.0 | 81.6 | 82.1 |

Table 9: Full experimental results of our models compared to three different backbones.

## H  MIND2WEB EXPERIMENT DETAILS

In the experiments on Mind2Web, we give the models a raw screenshot, a task description, and previous actions as inputs. Models are then prompted to directly predict the next action.

Screenshots from Mind2Web's original observation capture the entire page, which can go as long as over 8:1 in height-to-width ratio. Observations of this length disconnect from human observations. Similar to SeeClick (Cheng et al., 2024), we randomly crop around the ground truth element, with a random window size at least as big as those used in SeeClick.

We evaluated *CogAgent-Chat* in the same setting as a baseline, as *CogAgent-Chat* is the recommended version of CogAgent for agent application. We used the exact prompt in Hong et al. (2023) that is used for agent operation with grounding. To translate model output into GUI action, we convert the model's predicted bounding box into a point of operation on the web page by taking the center point of the bounding box. Here is an example prompt we used:

---

**Prompt example for CogAgent**

What's the proper procedure for "Add to cart one Private Vehicle Pass for Yosemite National Park for April 30, for License Plate number 12345 and Zip Code 94587
Previous Actions:
```
[div]  All About Passes -> CLICK
[input]   -> TYPE: Yosemite
[link]  Yosemite National Park id: 74296 -> CLICK
[img]  0 Private Vehicle Pass -> CLICK
[textbox]  Zip Code(Required) -> TYPE: 94587 "
```
(with grounding).

---

# I ABLATION ON TASK TYPES

The tasks defined in Section 2 are categorized into QA, Caption, OCR, and Grounding. To evaluate the impact of different types of training samples on downstream performance, we conducted a comprehensive ablation study. For instance, we trained a captioning model using a combination of Caption samples from the MultiUI dataset and LLaVA data, and analyzed its performance on various downstream benchmarks. Figure 6 presents the results of this study, highlighting several key findings. Specifically, both QA and Caption samples significantly enhance performance in GUI understanding (VisualWebBench) and OCR-related benchmarks (DocVQA), while OCR samples primarily benefit tasks in GUI understanding (VisualWebBench) that require basic text recognition. Notably, relying solely on grounding samples derived from web data can adversely affect performance on the RefCOCO+ benchmark, as it involves natural scene images rather than webpage-like images. Overall, incorporating all sample types yields the most balanced performance across all four benchmark categories.

## J   PROMPT VARIATION

To improve the generalizability of the generated conversation samples, we diversify the instruction templates for each task. First, we create a detailed task description and initial example templates. Then, using GPT-4o, we generate 200 varied prompt templates based on the task descriptions and human-provided demonstrations.

# K COMPARISON WITH OTHER DATASETS

| Datasets | Task Types | #Samples |
|---|---|---|
| SeeClick | Grounding, Captioning | 1M |
| Ferret-UI | QA, Grounding, Captioning | 841K |
| Mind2Web | Web agent | 2K |
| MultiUI | QA, OCR, Grounding, Captioning, Action Prediction | 7.3M |

Table 10: Comparision among GUI-related datasets.

