# OpenReview forum: "Harnessing Webpage UIs for Text-Rich Visual Understanding"
_ICLR.cc/2025/Conference — ICLR 2025 Poster_

### Official Review · Reviewer_oGMe · 2024-10-29

**Soundness:** 4
**Presentation:** 3
**Contribution:** 3
**Rating:** 6
**Confidence:** 5

**Summary:**

This paper presents MultiUI, a new dataset that leverages webpage user interfaces as a structured and diverse multimodal data source for training large language models to understand text-rich visual content. The dataset consists of 7.3 million samples, categorized across various tasks such as visual understanding, text recognition, and GUI grounding. The authors address challenges of noise, hallucination, and generalization limitations in existing multimodal data sources. The paper introduces a two-stage training strategy, achieving significant gains in GUI-related tasks and general multimodal understanding. MultiUI demonstrates potential for improving MLLMs' capabilities in both GUI-specific and broader text-rich visual tasks, highlighting structured web data as a valuable training resource.

**Strengths:**

1. **Useful Dataset on a Good Problem**: The paper tackles the challenging and impactful task of text-rich visual understanding by introducing MultiUI, a large, well-structured dataset derived from webpage UIs. This dataset fills a critical gap by leveraging the naturally organized and text-embedded nature of UIs, enhancing MLLMs’ abilities to interpret complex, multimodal information in real-world scenarios.

2. **Comprehensive Experiment Results**: The authors present thorough experiments that validate MultiUI’s effectiveness across GUI-related and general multimodal benchmarks, demonstrating significant performance improvements and robust generalization capabilities. The results solidly support MultiUI’s utility and showcase its potential in both domain-specific and cross-domain applications.

**Weaknesses:**

One notable weakness of the paper is the limited discussion around the quality of the generated data and labels within MultiUI. While the dataset construction relies on accessibility trees and LLM-generated instructions to filter and label data, the paper lacks in-depth analysis or validation of data quality, particularly concerning the accuracy, consistency, and noise levels in generated labels. Without a rigorous assessment or benchmarks comparing the quality of MultiUI labels against human-labeled or gold-standard datasets, it is difficult to gauge the dataset’s reliability for downstream tasks, potentially impacting the generalizability and robustness of models trained on it.

**Questions:**

In the data construction process, how do you ensure the quality of the data points?

---

> ### Author Response · Authors · 2024-11-26
>
> We appreciate the **constructive feedback** and have carefully reviewed the concerns raised about the **quality assessment** of the generated data and labels in the **MultiUI dataset**. Below, we provide **clarifications** and **improvements** to address these points.
>
> ---
>
> ## Discussions of Data Quality
>
> To ensure the **quality** of the generated data and labels in **MultiUI**, we conducted a **human study** on 100 randomly selected samples. The results indicate that:
>
> - **97% of the samples are accurate**, with only **3% containing position errors**, which we believe do not significantly impact overall data quality.
> - Potential issues such as **hallucinations** or **mislabeling** may exist; however, these are mitigated by the robust **construction pipeline** leveraging **accessibility trees** and **LLM-generated instructions** for filtering and labeling.
>
> More importantly, **the experimental results on 19 downstream tasks strongly demonstrate the dataset's effectiveness**. Models trained on **MultiUI** achieved **significant performance gains** across diverse benchmarks. This highlights the **reliability** of MultiUI for downstream tasks and confirms its **practical utility** despite any minor imperfections.
>
> ---
>
> ## Assurance of Data Quality in Construction
>
> Our pipeline includes **multiple safeguards** to ensure the quality of data points:
>
> 1. **Accessibility Trees**: Leveraging accessibility trees significantly reduced **noise** compared to traditional **rule-based** or raw **HTML parsing** approaches.
> 2. **Multi-Step Curation Process**: Content filtering is conducted via the **Llama-3-70B-Instruct model**, ensuring only **valid** and **contextually relevant** data is included.
> 3. **Task-Specific Heuristics**: Heuristics applied during task extraction eliminated **ambiguous** or **low-information samples**.
> 4. **Downstream Task Performance**: To verify alignment, we tested a range of models fine-tuned on **MultiUI** and observed **consistent performance improvements** across benchmarks.

---

> > ### Author Response · Authors · 2024-11-26
> > **Response to Ethics Review Flags**
> >
> > We acknowledge the importance of addressing potential **ethical implications** in constructing and utilizing the **MultiUI dataset**. Below, we address each flagged concern to demonstrate our compliance with **ethical standards**:
> >
> > ### 1. Privacy, Security, and Safety
> > - **Dataset Sources**: The dataset exclusively uses **publicly available web data URLs** (e.g., **Huggingface FineWeb**). Significant precautions were taken to ensure **no private or sensitive information** is included.
> > - **Filtering Mechanisms**: A robust filtering pipeline, leveraging models like **Llama-3-70B-Instruct**, removes **inappropriate**, **harmful**, or **sensitive content** (e.g., adult material, gambling, discriminatory language). Any **URLs** or data related to **identifiable personal information** are excluded.
> > - **Annotations**: All annotations are generated through **automated processes** using **LLMs** and **accessibility trees**. **No human annotators** interacted with potentially sensitive raw data, safeguarding privacy.
> >
> > ### 2. Legal Compliance (e.g., GDPR, Copyright, Terms of Use)
> > - **GDPR Compliance**: The dataset aligns with **GDPR principles** by avoiding the processing of **personal data**. It focuses exclusively on **publicly accessible web pages** that do not require login or user interaction.
> > - **Copyright Considerations**: Screenshots included comply with **fair use principles**, as the dataset is intended solely for **research** and **educational purposes**.
> > - **Terms of Use**: Ethical web scraping practices were followed, such as respecting **robots.txt restrictions**, to avoid conflicts with **website terms of service**.

---

> ### Comment · Reviewer_oGMe · 2024-12-03
> **Response to the author's rebuttal**
>
> Thanks for your response. It resolves most of my concerns. As a result, I will keep my positive rating and increase my confidence score.

---

> > ### Author Response · Authors · 2024-12-03
> > **Thank you for keeping positive rating and the high confidence score!**
> >
> > Thank you for your positive feedback! We are glad that our response addressed your concerns. We really appreciate your increased confidence score and positive rating!
> >
> >
> > Best,
> >
> > authors

---

### Official Review · Reviewer_8zAm · 2024-10-30

**Soundness:** 3
**Presentation:** 4
**Contribution:** 4
**Rating:** 8
**Confidence:** 4

**Summary:**

This work introduces MultiUI, a dataset of 7.3 million VQA examples spanning 9 types of tasks covering topics of perception, comprehension, grounding, and reasoning of webpage UIs, rendered on both desktop and mobile platforms.

Specifically, the authors collect data leveraging information from the websites both directly visible from the rendered view as well as the originating accessibility tree and DOM structure and generate task data using a mixture of rules and various foundation models:
1. Webpage captioning: LLama 3 70B summarizes the website based on the accessibility tree.
2. Webpage QA: Llama 3 70B generates questions and answers for a given website based on the accessibility tree.
3. Embedded image captioning: GPT-4o-mini generates a caption of an image on the website, including context provided by the accessibility tree.
4. Embedded image QA: Llama 3 70B generates questions and answers for an embedded image, based on the generated caption.
5. Action prediction: PlayWright is used to interact with elements on the website to generate multi-choice Q&A pairs. The task is predicting the title of a target page when interacting with an element on the original website.
6. Element OCR: the task is to predict the text in an area highlighted with a red bounding box (set-of-mark prompting) in the rendered view. Relevant elements are selected based on minimum text length in the DOM.
7: Heading OCR:  the task is to predict the title of the website.
8: Action Grounding: Llama 3 70B generates request for click targets to execute a certain action, based on information from the accessibility tree. Data is generated for both multiple-choice and direct answer settings.
9: Element Grounding: coordinates and a textual descriptions are extracted from a DOM tree, with the task to predict the coordinates based on the description. Again both multi-choice and direct answer settings are considered.

The show the efficacy or the collected data to improve performance on GUI understanding and grounding, the authors train multimodal LLMs using various LLM backbones and show consistent and significant improvements over models trained on LLaVA 1.5 or LLaVA-NeXT data mixtures when incorporating their proposed MultiUI data.

**Strengths:**

* The collected dataset appears relevant to the burgeoning field of LLM based UI understanding and control.
* The method introduced to collect the dataset is described clearly and the identified sub-tasks are relevant and well motivated.
* The models fine tuned on the introduced data achieve strong results across benchmarks, which supports the utility of the data collected.

**Weaknesses:**

* The description of the grounding task data generation (2.3.3) could be more specific. The description of how Llama 3 is used is not very clear to me ("[the model] is not only prompted to generate multiple grounding instructios to predict the bounding box of a given element but also provides the corresponding ground-truth bounding boxes"). For element grounding the extraction from the DOM tree is not described. Is the "element description" simply the text corresponding to an element (as per figure 15)?
* While mentioned in the baseline section (3.3) and measured on general OCR and grounding benchmarks, GPT-4o is not considered for GUI understanding and grounding (table 2). Given that GPT-4o shows significantly stronger results on the general benchmarks shown in table 3, it would appear to be a stronger / more relevant baseline to also consider in table 2. This is particularly noteworthy since section 4.1 discusses GPT-4V as the "most advanced MLLM" in the comparison.
* Minor: figure 4 and 6 rely on color, hard to follow on a black & white print.

**Questions:**

* The ablation on task type (section 4.7) shows that RefCOCO+ performance deteriorates slightly when adding MultiUI Grounding data, but seems to meaningfully improve when adding other data, particularly QA data. The discussion attributes the reduction in performance to domain shift but doesn't offer suggestions why MultiUI as a whole may still benefit RefCOCO+. Is there some understanding why this may be the case?
* ScreenSpot results are not reported for e.g. SeeClick or CogAgent, but as far as I can tell they are reported, e.g. here: https://github.com/njucckevin/SeeClick Can they be added to table 2?
* In section 3.1 the high-resolution strategy is briefly introduced. However, the exact tiling setup is not iterated on, including in appendix F. Appendix F lists a "maximum resolution" but I presume this is the resolution of an individual tile? Since the exact tiling strategy has significant impact on some of the reported benchmarks (such as DocVQA), I'd recommend to add a few more details to clarify the implementation and facilitate reproducibility in appendix F.
* For test sets like MMMU, RefCOCO+, etc. that have multiple splits that are somewhat widely used, please clarify which sets are used (perhaps in appendix G).

---

> ### Author Response · Authors · 2024-11-26
>
> Thank you for your thoughtful review. We’re glad you found our **dataset relevant**, our **methodology clear**, and our **results strong**!
>
> ### Construction Process of Grounding Task Samples
> - **Action grounding task**:
>   Given the **accessibility tree** of a webpage where each **UI element** is accompanied by its **bounding box**, **Llama** is instructed to:
>   1. Select an element (e.g., a button to navigate to the next page) on the webpage.
>   2. Generate an **instruction** (e.g., "change to next page") that targets the selected element.
>   The resulting `<instruction, bounding box>` pair is collected as a sample for the **action grounding task**.
>   The **prompt used for Llama** is detailed in **Appendix C**.
>
> - **Element grounding task**:
>   - For **text elements** (e.g., `<p>`, `<h>` tags), **element descriptions** are their **inner text** within the HTML.
>   - For **icon/button elements**, descriptions are extracted from the **“aria-text” attribute** of the corresponding tag.
>
> ---
>
> ### Adding GPT-4o Performance in Table 2
> We have updated **Table 2** to include **GPT-4o**.
> - Notably, **GPT-4o** only shows a **marginal improvement** over **GPT-4V** on the **GUI grounding benchmark** (**ScreenSpot**).
>
> ---
>
> ### View in Color
> Thank you for the suggestion! We have updated all figures for **better color visualization**.
>
> ---
>
> ## Ablation Study of Grounding Data
> - **Table 1** highlights that **MultiUI** includes **526K embedded image captions & QA data**, where a significant portion of the embedded images are **natural images** (e.g., photos of landscapes, buildings, animals, and plants).
> - This dataset **enhances the model’s understanding** of **natural images**, leading to improved performance on **RefCOCO+**.
> - This analysis highlights the **gap between GUI/Doc/Chart/OCR images** and **natural images**, which we have explicitly discussed in the updated version.
>
> ---
>
> ### ScreenSpot Results of SeeClick and CogAgent
> Thank you for the reminder! We have added these results to **Table 2**:
> - Our **UIX-Qwen2-7B** **outperforms** both **SeeClick** and **CogAgent**, despite being **smaller** in model size (**7B vs. SeeClick: 9.6B, CogAgent: 18B**).
>
> ---
>
> ### Implementation of High-Resolution Strategy
> - The **"maximum resolution" column** in **Appendix E** refers to the **total resolution** of multiple tiles.
> - We **follow the dynamic resolution strategy** described in **LLaVA-NeXT** [1].
>
> ---
>
> ### Test Splits
> We have added the **test splits** for each benchmark in **Appendix G** of the revised version.
>
> ---
>
> [1] [LLaVA-NeXT Blog (2024-01-30)](https://llava-vl.github.io/blog/2024-01-30-llava-next/)

---

> > ### Comment · Reviewer_8zAm · 2024-11-27
> >
> > I want to thank the authors for all the helpful clarifications and changes to the draft. I will maintain my rating of 8: accept, good paper.

---

> ### Author Response · Authors · 2024-11-30
> **Thank you!**
>
> Thank you for maintaining your positive assessment of our work! We sincerely express our gratitude for your constructive comments and suggestions!
>
> Best, Authors

---

### Official Review · Reviewer_exHC · 2024-11-03

**Soundness:** 2
**Presentation:** 3
**Contribution:** 2
**Rating:** 5
**Confidence:** 3

**Summary:**

This paper addresses the challenge of improving text-rich visual understanding in multimodal large language models, focusing on the ability to interpret both textual and visual elements. MultiUI aims to improve MLLM performance in document understanding, GUI comprehension, and grounding, enhancing generalization across web and other domains. Experimental results show that models trained on MultiUI significantly outperform baselines in tasks such as GUI understanding, OCR, and grounding.

**Strengths:**

1. The introduction of MultiUI as a large, diverse dataset specifically designed to enhance multimodal understanding using structured web UI data.
2. The paper demonstrates the dataset's impact, showing substantial performance gains over existing baselines across various multimodal tasks, emphasizing its importance for text-rich visual understanding.

**Weaknesses:**

1. There's a lack of documentation, making the dataset hard to navigate and providing no usage guide for the code. The authors should improve accessibility and include a clear code guide.

2. In the construction pipeline of MultiUI, while the use of Llama and LLaVA may be reasonable for certain tasks, relying on them for higher-level tasks like question answering raises concerns about reliability. Using these models for complex tasks in the dataset risks introducing biases and undermining the robustness of the evaluation.

**Questions:**

Recommended to clarify whether any human validation was conducted on the MultiUI dataset, including the QA tasks generated within the dataset.

---

> ### Author Response · Authors · 2024-11-26
>
> Thank you for recognizing **MultiUI** as a diverse dataset for enhancing **multimodal understanding** and its significant impact on improving performance across various **text-rich visual tasks**. We appreciate your acknowledgment of its importance in advancing this area!
>
> ---
>
> ### Documentation and Code Guide
>
> We have provided an **anonymous GitHub repository** linked on the first page, which includes:
>
> - **README**
> - **Example data**
> - **Training & evaluation code**
>
> In response to the reviewer's suggestion, we have added:
>
> - **More detailed documentation** for the example data.
> - A **clearer code guide**.
>
> **Thank you for the valuable suggestion!**
>
> ---
>
> ### Relying on Llama and LLaVA
>
> - As shown in **Figure 2**, the construction pipeline of **MultiUI** does **not** rely on **LLaVA**.
> - To evaluate the effectiveness of **MultiUI**, we conducted experiments across **19 diverse benchmarks**, covering:
>   - **GUI-related**
>   - **OCR/Doc/Chart**
>   - **General domains**
>
> Detailed results are provided in: **Table 2**, **Table 3**, **Table 4**, and **Table 5**.
>
> These **comprehensive evaluations** robustly demonstrate the **effectiveness** and **versatility** of models trained on **MultiUI**.
>
> To further demonstrate the quality of our dataset:
>
> - We conducted a **human study** on **100 randomly selected samples** from **MultiUI**.
> - A minor issue was identified: **3 samples** contained **position errors**.
> - However, this **minor issue** does **not significantly affect** the overall **data quality**.
>
> Thank you again for your thoughtful feedback, which has helped us further improve the clarity and presentation of our work!

---

> > ### Comment · Reviewer_exHC · 2024-12-03
> >
> > Thank you for your response. However, the key point is that LLMs play a very significant role in the data curation process. Additionally, since there hasn't been sufficient verification of this, concerns about dataset quality remain unresolved.

---

> > > ### Author Response · Authors · 2024-12-03
> > >
> > > Thank you for your response.
> > >
> > > In terms of the data quality, the experimental results on 19 benchmarks could demonstrate the good quality and effectiveness of MultiUI across tasks (Table 2,3,4,5).
> > >
> > > Furthermore, we conducted another more comprehensive human study following up the previous one reported in our last response. This human study has a data size of 450 and includes detailed breakdown analysis of each task type (50 samples for each task type). The table below presents the results. The total error rate (13/450=2.89%) still remains quite low, further demonstrating the robustness of our data.
> > > |                  | Web Capt. | Web QA | Img QA | Img Capt. | Head. OCR | Elem. OCR | Act. Pred | Elem. Ground | Act. Ground | total |
> > > |------------------|------------|------------|----------|---------------|-------------|-------------|-------------------|----------------|---------------|-------|
> > > | **error count**  | 2          | 2          | 1        | 1             | 0           | 2           | 0                 | 3              | 2             | 13    |
> > > ||
> > >
> > > Best,
> > >
> > > Authors

---

### Official Review · Reviewer_K6jg · 2024-11-04

**Soundness:** 3
**Presentation:** 2
**Contribution:** 2
**Rating:** 6
**Confidence:** 4

**Summary:**

This paper introduces MultiUI, a large-scale dataset of website UIs annotated for diverse UI-related tasks, including captioning, visual question answering, OCR, and UI interaction grounding. MultiUI leverages web screenshots and accessibility trees as key components of its data pipeline, with annotations synthesized using powerful LLMs to support these specific UI tasks. The second part of the paper explores training Vision-Language Models (VLMs) on this new dataset and evaluates their performance on GUI understanding and grounding benchmarks.

**Strengths:**

- The paper is easy to follow and provides substantial figures for improving its understanding.
- The methodology for defining the data collection pipeline seems valid and the qualitative examples show that the samples are relevant.
- The proposed dataset provides substantial annotations relevant for the UI understanding field. It is a large-scale dataset, and can be relevant for the VLM community.
- Experimental results of training Llava-based VLMs on the proposed data shows that models performance improve with respect to models not trained on it.

**Weaknesses:**

- The paper lacks a comprehensive comparison with other datasets in the field, in terms of number of samples, and types of annotations and tasks they can perform. Some examples are SeeClick, Ferret-UI, Mind2Web, or WebArena, among others.
- The selection of baselines seems limited, mostly relying on Llava. Most of the results on other baselines and architecture appear empty (referring to tables 2 and 3). Authors need to thoroughly evaluate closed models like GPT4o or Claude 3.5 in the proposed benchmarks in order to offer a more strong comparison, alongside powerful open-source options like MolMo, Gemini, Llama3.2.
- Authors only present finetuning experiments on Llava-based models, lacking performing experiments on other baseline architectures to show the benefits of the proposed dataset.
- While the paper presents contributions in annotating tasks across visual understanding and reasoning, OCR, and UI grounding, the novelty of its approach is limited compared to recent advancements in the field, such as SeeClick, Ferret-UI, Mind2Web, and WebArena. The primary distinction appears to be in the scale of annotation rather than introducing fundamentally new methodologies or task definitions.
- Authors claim that the pipeline solves hallucination in the annotation process, a phenomena observed in previous works, but the paper does not describe how exactly this hallucination prevention is done.
- The experiments section is missing a batch of experiments analyzing what is the performance of models when trained on other similar competitor datasets in the field.

**Questions:**

- How did the authors filter websites for making sure it does not contain harmful or adult content? They provide some detail on LLM based filtering on section 2.2, but this can be insufficient. Did you consider human inspection to enhance this process?
- How does this dataset compare with other datasets in the field? A statistical comparison on the quantity and quality of samples is missing. Also, in the experiments section, we are missing a comparison on how VLMs finetuned on other dataset perform on the proposed benchmarks, in order to see the benefits of the proposed data in performance terms.
- How did authors solve hallucination when creating this dataset synthetically using LLMs?
- Have authors analyzed the train/test overlap, to avoid data contamination from train to test?
- Have authors considered finetuning other VLMs than Llava?
How does image resolution impact performance on the proposed benchmarks? Are there additional patterns observed in the experiments when comparing different Llava variants? A discussion of these insights would be appreciated.

(see the weaknesses section for additional questions)

---

> ### Author Response · Authors · 2024-11-26
> **Response (1/2)**
>
> Thank you for your thoughtful review and for recognizing the **strengths of our work**, including the **clarity** of our paper, the **validity** of our data pipeline, and the **value of the MultiUI dataset** for the community. We appreciate your positive feedback on our contributions and **experimental results**.
>
> ---
>
> ### Comparison with Other Datasets
>
> We have added a table to compare the proposed **MultiUI** with some relevant datasets in **Appendix K**. We can see that **MultiUI** not only covers **most task types**, including **QA, OCR, grounding, captioning, and action prediction**, but also contains **most training samples**. This highlights the **broad applicability** of the proposed MultiUI data for advancing **text-rich visual understanding** and **grounding across various scenarios**.
>
> | **Datasets** | **Task Types**                              | **#Samples** |
> |--------------|---------------------------------------------|--------------|
> | SeeClick     | Grounding, Captioning                      | 1M           |
> | Ferret-UI    | QA, Grounding, Captioning                  | 841K         |
> | Mind2Web     | Web agent                                  | 2K           |
> | MultiUI      | QA, OCR, Grounding, Captioning, Action Prediction | 7.3M       |
>
> *Table: Comparison among GUI-related datasets.*
>
> ---
>
> ### More Baseline Results (e.g., GPT4o or Claude 3.5, MolMo, Gemini, Llama3.2)
>
> We have added results of **GPT4o** and **Gemini 1.5 Pro** into **Table 2** and **Table 3** in the revision version. We will also include results for **Molmo** and **Llama 3.2** in the revised version.
>
> ---
>
> ### Experiment Results on Other Backbone Architectures
>
> Even though we adopted the **LLaVA architecture**, we explored **different variants of the LLaVA models**. Specifically, we experimented with:
>
> 1. **Various base language models** including **Vicuna, Llama, and Qwen**.
> 2. **Different vision encoders** such as **CLIP** and **SigLIP**.
> 3. **Two versions** of general visual instruction-tuning data: **LLaVA-1.5 data** and **LLaVA-NeXT data**.
>
> As shown in **Table 2** and **Table 3**, the proposed **MultiUI dataset consistently leads to significant improvements** across all LLaVA model variants.
>
> ---
>
> ### Novelty
>
> The contribution of our work is not only about **scaling up**. We propose to harvest **diverse multimodal instructions** of **multiple task types** from **webpage UIs** with the aid of text-based **LLMs**, which is not without challenges, such as:
>
> - Handling **lengthy raw HTML**.
> - Defining **generalizable tasks** from the web domain to other domains.
> - Ensuring the **diversity** of synthesized instructions.
>
> ---
>
> ### Alleviating Hallucination
>
> We neither claim nor aim to **completely solve hallucination**, but we aim to **alleviate it**. Unlike vision-only models that rely on images, our method leverages **HTML**, which is:
>
> 1. **Carefully written by human engineers/designers** and corresponds **precisely** to the rendered webpage.
> 2. Processed into an **accessibility axtree**, containing no **hallucinated elements** theoretically.
>
> However, we acknowledge that **advanced text models** may still exhibit some **hallucination phenomena** during data construction.
>
> ---

---

> ### Author Response · Authors · 2024-11-26
> **Response (2/2)**
>
> ### Fine-Tuning Experiments on Similar Datasets
>
> We considered relevant datasets for specific scenarios like **document understanding** (e.g., **DocVQA, ChartQA**) and **GUI-related tasks** (e.g., **SeeClick, CogAgent**). However:
>
> - To our knowledge, **MultiUI** is the **first automatically created dataset** constructed from **web UIs**, benefitting **diverse scenarios**.
> - While we didn’t directly fine-tune models on these datasets due to **limited computing costs**, we compared our trained models with:
>
>   1. **Document understanding models** (e.g., **DocOwl-1.5, DocOwl-2** in **Table 3**).
>   2. **GUI grounding models** (e.g., **SeeClick, CogAgent** in **Table 2**).
>
> We found that **UIX-Qwen2** **outperformed specialized models** on both **document-related** (e.g., **DocVQA, ChartQA**) and **GUI-related** (e.g., **VisualWebBench, ScreenSpot**) benchmarks.
>
> ---
>
> ### Quality of Website Curation
>
> The employed URLs are taken from **FineWeb** [1], which involves:
>
> - **Filtering out malicious/NSFW websites**.
> - Additional filtering based on the **accessibility tree**, which simplifies webpage content.
>
> We conducted a **human inspection** on 300 randomly selected samples and found no objectionable content, such as **adult material**, **gambling**, or **violence**.
>
> ---
>
> ### Data Decontamination
>
> We considered **data decontamination** during the construction process. As described in **Section 2.1**, we:
>
> - Removed all URLs appearing in **downstream benchmarks** (ratio: **<1%**).
>
> ---
>
> ### Discussions
>
> #### Impact of Image Resolution
>
> Webpage images typically have **higher resolutions** (e.g., **1280 × 1280**) compared to natural images (e.g., COCO). We hypothesize that **higher resolutions yield better performance** on web UIs. Our experiments with the **LLaVA-1.5 architecture** revealed:
>
> - **Smaller resolutions (336 × 336)** hinder performance due to loss of detail.
> - **Doubling resolution (672 × 672)** led to **notable improvements** while balancing training costs.
>
> #### Comparison Among Different LLaVA Variants
>
> As described in **Section 3.1** and **Appendix E**, we conducted experiments on three LLaVA variants:
>
> 1. **UIX-Vicuna**
> 2. **UIX-Llama3.1**
> 3. **UIX-Qwen2**
>
> Key findings:
>
> - Training **Llama3.1** was **unstable**.
> - **Qwen2** was used as the primary base LM.
> - Results varied based on **training data** (e.g., improvements on **DocVQA** were more significant with **LLaVA-1.5 data** than with **LLaVA-NeXT data**).
>
> ---
>
> ### References
>
> [1] **The FineWeb Datasets: Decanting the Web for the Finest Text Data at Scale**.

---

> > ### Comment · Reviewer_K6jg · 2024-12-02
> >
> > Thanks for spending the time performing experiments and adding clarity to the paper. I appreciate the experiment with Llava on the impact of image resolution, it follows the expected behavior. I also appreciate experiments done with GPT4o and Llama3.2. However, there are still many blank cells in Tables 2 and 3. Claude's results are missing. I recommend spending time adding these remaining scores.
> >
> > Thanks for clarifying the other concerns, they are correctly addressed.
> >
> > Taking this into account, I will maintain my original score of 6.

---

> ### Author Response · Authors · 2024-12-02
> **Thank you for maintaining the positive rating!**
>
> Thank you for maintaining your positive rating and for appreciating the additional experiments. We are glad that these efforts have clarified your concerns. We will add your mentioned results in the revised version.
>
> Best, authors

---

### Author Response · Authors · 2024-11-26

- In terms of **writing**:
  - We have significantly improved our **writing quality**.
  - The quality of **figures 1-16** has been greatly improved.
- In terms of experiment **results and numbers**:
  - We added the result of **GPT-4o** and **Gemini 1.5 Pro** in **table 2 and table 3**, and the result of **LLaVA-Qwen2-M2W** in **table 4** to offer **stronger baseline comparisons**.
- We added a **test split column** for each benchmark in **Appendix G** to further clarify our benchmark setting.
- **Appendix K** is added to include comparisons of **MultiUI against other similar datasets**, which highlights the diversity of MultiUI as a dataset.
- Changes to **Appendix E, I, J** are made as a result of improved writing.

---

### Meta-Review · Area_Chair_dfoZ · 2024-12-18

**Metareview:**

This paper presents MultiUI, a large-scale dataset derived from website UIs designed to improve multimodal learning in text-rich environments. By integrating rendered screenshots, accessibility trees, and LLM-assisted annotations, MultiUI covers a range of UI-related tasks including captioning, VQA, OCR, and element grounding. The dataset’s breadth and structured data collection methodology address known challenges in text-heavy visual understanding, and experiments show that training with MultiUI improves performance on various GUI benchmarks and general multimodal tasks.
Reviewers appreciated the clarity of the dataset construction and the potential impact on VLM research. they highlighted several points for improvement, comparing MultiUI more thoroughly to similar datasets, evaluating more diverse baselines beyond LLaVA-based models, verifying data quality and annotation reliability, and clarifying certain aspects of the grounding task. Despite these limitations, MultiUI scale, relevance, and demonstrated utility make it a valuable resource. I recommend acceptance.

**Additional Comments On Reviewer Discussion:**

During the discussion, the authors offered clarifications and their plans for documenting the dataset. Although some reviewers wished for more comparisons and stronger baselines, the author's responsiveness and willingness to refine their methodology were well-received. The general consensus is that MultiUI represents a meaningful step forward in supporting capabilities in text-heavy visual scenarios, justifying acceptance.

---

### Decision · Program_Chairs · 2025-01-22

Accept (Poster)